# Estimating SARS-CoV-2 seroprevalence and epidemiological parameters with uncertainty from serological surveys

Daniel B Larremore[1,2]\*, Bailey K Fosdick[3], Kate M Bubar[4,5], Sam Zhang[4], Stephen M Kissler[6], C Jessica E Metcalf[7], Caroline O Buckee[8,9], Yonatan H Grad[6]\*

[1]Department of Computer Science, University of Colorado Boulder, Boulder, United States; [2]BioFrontiers Institute, University of Colorado Boulder, Boulder, United States; [3]Department of Statistics, Colorado State University, Fort Collins, United States; [4]Department of Applied Mathematics, University of Colorado Boulder, Boulder, United States; [5]IQ Biology Program, University of Colorado Boulder, Boulder, United States; [6]Department of Immunology and Infectious Diseases, Harvard T.H. Chan School of Public Health, Boston, United States; [7]Department of Ecology and Evolutionary Biology and the Woodrow Wilson School, Princeton University, Princeton, United States; [8]Department of Epidemiology, Harvard T.H. Chan School of Public Health, Boston, United States; [9]Center for Communicable Disease Dynamics, Harvard T.H. Chan School of Public Health, Boston, United States

\*For correspondence:
daniel.larremore@colorado.edu (DBL);
ygrad@hsph.harvard.edu (YHG)

**Competing interests:** The authors declare that no competing interests exist.

**Abstract** Establishing how many people have been infected by SARS-CoV-2 remains an urgent priority for controlling the COVID-19 pandemic. Serological tests that identify past infection can be used to estimate cumulative incidence, but the relative accuracy and robustness of various sampling strategies have been unclear. We developed a flexible framework that integrates uncertainty from test characteristics, sample size, and heterogeneity in seroprevalence across subpopulations to compare estimates from sampling schemes. Using the same framework and making the assumption that seropositivity indicates immune protection, we propagated estimates and uncertainty through dynamical models to assess uncertainty in the epidemiological parameters needed to evaluate public health interventions and found that sampling schemes informed by demographics and contact networks outperform uniform sampling. The framework can be adapted to optimize serosurvey design given test characteristics and capacity, population demography, sampling strategy, and modeling approach, and can be tailored to support decision-making around introducing or removing interventions.

## Introduction

Serological testing is a critical component of the response to COVID-19 as well as to future epidemics. Assessment of population seropositivity, a measure of the prevalence of individuals who have been infected in the past and developed antibodies to the virus, can address gaps in knowledge of the cumulative disease incidence. This is particularly important given inadequate viral diagnostic testing and incomplete understanding of the rates of mild and asymptomatic infections (*Sutton et al., 2020*). In this context, serological surveillance has the potential to provide information about the true number of infections, allowing for robust estimates of case and infection fatality rates (*Fontanet et al., 2020*) and for the parameterization of epidemiological models to evaluate the possible impacts of specific interventions and thus guide public health decision-making.

The proportion of the population that has been infected by, and recovered from, the coronavirus causing COVID-19 will be a critical measure to inform policies on a population level, including when and how social distancing interventions can be relaxed, and the prioritization of vaccines (*Bubar et al., 2021*). Individual serological testing may allow low-risk individuals to return to work, school, or university, contingent on the immune protection afforded by a measurable antibody response (*Weitz et al., 2020*; *Larremore, 2020*). At a population level, however, methods are urgently needed to design and interpret serological data based on testing of subpopulations, including convenience samples such as blood donors (*Valenti et al., 2020*; *Erikstrup et al., 2021*; *Fontanet et al., 2020*) and neonatal heel sticks, to reliably estimate population seroprevalence.

Three sources of uncertainty complicate efforts to learn population seroprevalence from subsampling. First, tests may have imperfect sensitivity and specificity, and studies that do not adjust for test imperfections will produce biased seroprevalence estimates. Complicating this issue is the fact that sensitivity and specificity are, themselves, estimated from data (*Larremore and Fosdick, 2020*; *Gelman and Carpenter, 2020*), which can lead to statistical confusion if uncertainty is not correctly propagated (*Bendavid et al., 2020*). Second, the population sampled will likely not be a representative random sample (*Bendavid et al., 2020*), especially in the first rounds of testing, when there is urgency to test using convenience samples and potentially limited serological testing capacity. Third, there is uncertainty inherent to any model-based forecast that uses the empirical estimation of seroprevalence, regardless of the quality of the test, in part because of the uncertain relationship between seropositivity and immunity (*Tan et al., 2020*; *Ward et al., 2020*).

A clear evidence-based guide to aid the design of serological studies is critical to policy makers and public health officials both for estimation of seroprevalence and forward-looking modeling efforts, particularly if serological positivity reflects immune protection. To address this need, we developed a framework that can be used to design and interpret cross-sectional serological studies, with applicability to SARS-CoV-2. Starting with results from a serological survey of a given size and age stratification, the framework incorporates the test's sensitivity and specificity and enables estimates of population seroprevalence that include uncertainty. These estimates can then be used in models of disease spread to calculate the effective reproductive number $R_{\text{eff}}$, the transmission potential of SARS-CoV-2 under partial immunity, forecast disease dynamics, and assess the impact of candidate public health and clinical interventions. Similarly, starting with a pre-specified tolerance for uncertainty in seroprevalence estimates, the framework can be used to optimize the sample size and allocation needed. This framework can be used in conjunction with any model, including ODE models (*Kissler et al., 2020a*; *Weitz et al., 2020*), agent-based simulations (*Ferguson et al., 2020*), or network simulations (*St-Onge et al., 2019*), and can be used to estimate $R_{\text{eff}}$ or to simulate transmission dynamics.

## Materials and methods

### Design and modeling framework

We developed a framework for the design and analysis of serosurveys in conjunction with epidemiological models (*Figure 1*), which can be used in two directions. In the forward direction, starting from serological data, one can estimate seroprevalence. While valuable on its own, seroprevalence can also be used as the input to an appropriate model to update forecasts or estimate the impacts of interventions. In the reverse direction, sample sizes can be calculated to yield seroprevalence estimates with a desired level of uncertainty and efficient sampling strategies can be developed based on prospective modeling tasks. The key methods include seroprevalence estimation, propagation of uncertainty through models, and model-informed sample size calculations.

### Bayesian inference of seroprevalence

To integrate uncertainty arising from test sensitivity and specificity, we used a Bayesian model to produce a posterior distribution of seroprevalence that incorporates uncertainty associated with a finite sample size (*Figure 1*, green annotations). We denote the posterior probability that the true population seroprevalence is equal to $\theta$, given test outcome data $X$ and test sensitivity and specificity characteristics, as $\Pr(\theta \mid X, \text{se}, \text{sp})$. Because sample size and outcomes are included in $X$, and because test sensitivity and specificity are included in the calculations, this posterior distribution over $\theta$

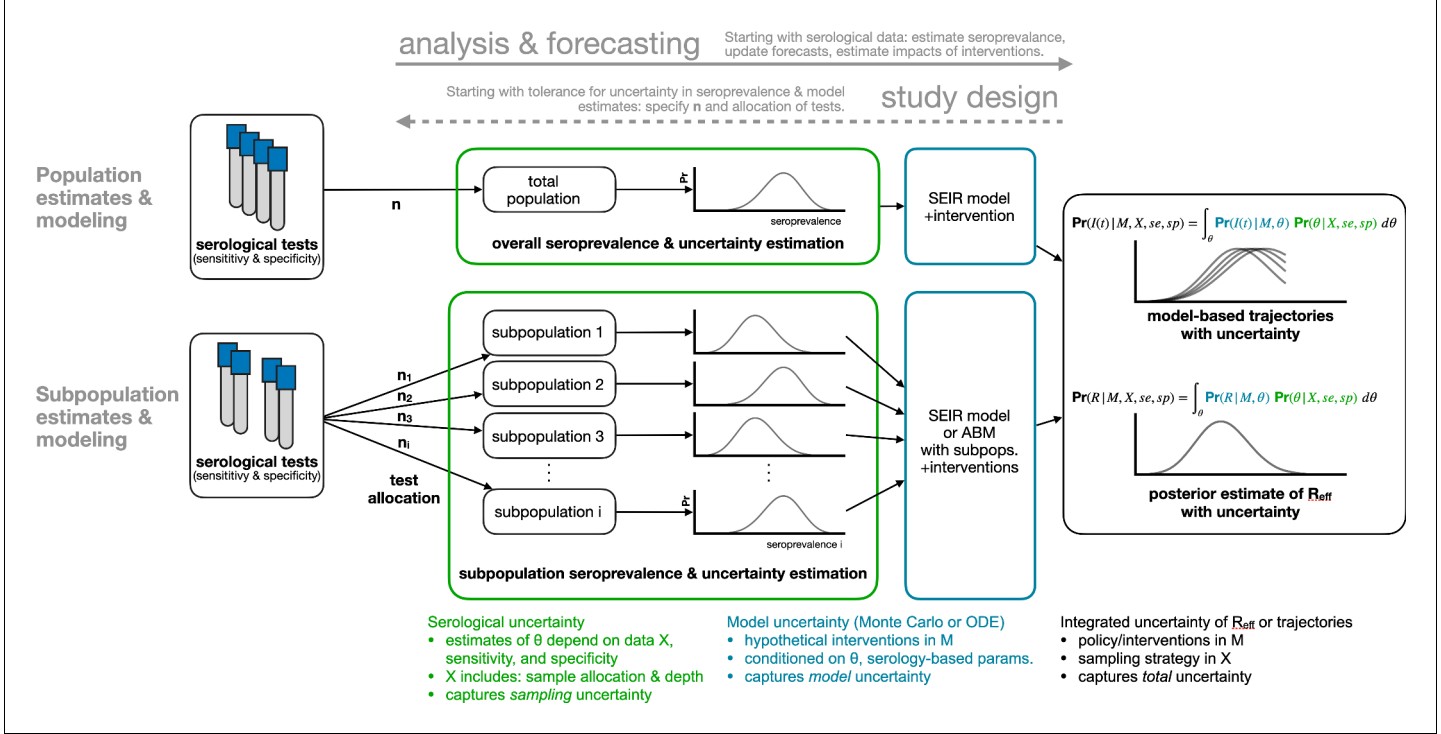

**Figure 1.** Framework for estimating seroprevalence and epidemiological parameters and the associated uncertainty, and for designing seroprevalence studies.

appropriately handles uncertainty due to limited sample sizes and an imperfect testing instrument, and can be used to produce a point estimate of seroprevalence or a posterior credible interval. The model and sampling algorithm are fully described in Appendix A1.

Sampling frameworks for seropositivity estimates are likely to be non-random and constrained to subpopulations. For example, convenience sampling (testing blood samples that were obtained for another purpose and are readily available) will often be the easiest and quickest data collection method (*Winter et al., 2018*). Two examples of such convenience samples are newborn heel stick dried blood spots, which contain maternal antibodies and thus reflect maternal exposure, and serum from blood donors (*Valenti et al., 2020*; *Erikstrup et al., 2021*; *Fontanet et al., 2020*). As a result, another source of statistical uncertainty comes from uneven sampling from a population.

To estimate seropositivity for all subpopulations based on a given sample (stratified, convenience, or otherwise), we specified a Bayesian hierarchical model that included a common prior distribution on subpopulation-specific seropositivities $\theta_i$ (Appendix A1). In effect, this allowed seropositivity estimates from individual subpopulations to inform each other while still taking into account subpopulation-specific testing outcomes. The joint posterior distribution of all subpopulation prevalences was sampled using Markov chain Monte Carlo (MCMC) methods (Appendix A1). Those samples, representing posterior seroprevalence estimates for individual subpopulations, were then combined in a demographically weighted average to obtain estimates of overall seroprevalence, a process commonly known as poststratification (*Little, 1993*; *Gelman and Carpenter, 2020*). We focus the demonstrations and analyses of our methods on age-based subpopulations due to their integration into POLYMOD-type age-structured models (*Mossong et al., 2008*; *Prem et al., 2017*), but note that our mathematical framework generalizes naturally to other definitions of subpopulations, including those defined by geography (*Fontanet et al., 2020*; *Nyc health testing data, 2020*; *Nisar et al., 2020*; *Malani et al., 2020*).

## Propagating serological uncertainty through models

In addition to estimating core epidemiological quantities (*Farrington and Whitaker, 2003*; *Farrington et al., 2001*; *Hens et al., 2012*) or mapping out patterns of outbreak risk (*Abrams et al.,*

*2014*), the posterior distribution of seroprevalence can be used as an input to any epidemiological model. Such models include the standard SEIR model, where the proportion seropositive may correspond to the recovered/immune compartment, as well as more complex frameworks such as an age-structured SEIR model incorporating interventions like school closures and social distancing (*Davies et al., 2020*; *Figure 1*, blue annotations). We integrated and propagated uncertainty in the posterior estimates of seroprevalence and uncertainty in model dynamics or parameters using Monte Carlo sampling to produce a posterior distribution of epidemic trajectories or key epidemiological parameter estimates (*Figure 1*, black annotations).

## Single-population SEIR model with social distancing and serology

To integrate inferred seroprevalence with uncertainty into a single-population SEIR model, we created an ensemble of SEIR model trajectories by repeatedly running simulations whose initial conditions were drawn from the seroprevalence posterior distribution. In particular, the seroprevalence posterior distribution was sampled, and each sample $\theta$ was used to inform the fraction of the population initially placed into the 'recovered' compartment of the model. Thus, uncertainty in posterior seroprevalence was propagated through model outcomes, which were measured as epidemic peak timing and peak height. Social distancing was modeled by decreasing the contact rate between susceptible and infected model compartments. A full description of the model and its parameters can be found in Appendix A2 and *Supplementary file 1*.

## Age-structured SEIR model with serology

To integrate inferred seroprevalence with uncertainty into an age-structured SEIR model, we considered a model with 16 age bins ($0 - 4, 5 - 9, \ldots 75 - 79$). This model was parameterized using country-specific age-contact patterns (*Mossong et al., 2008*; *Prem et al., 2017*) and COVID-19 parameter estimates (*Davies et al., 2020*). The model, due to *Davies et al., 2020*, includes age-specific clinical fractions and varying durations of preclinical, clinical, and subclinical infectiousness, as well as a decreased infectiousness for subclinical cases. A full description of the model and its parameters can be found in Appendix A2 and *Supplementary file 1*.

As in the single-population SEIR model, seroprevalence with uncertainty was integrated into the age-structured model by drawing samples from seroprevalence posterior to specify the fraction of each subpopulation placed into 'recovered' compartments. Posterior samples were drawn from the age-stratified joint posterior distribution whose subpopulations matched the model's subpopulations. For each set of posterior samples, the effective reproduction number $R_{\text{eff}}$ was computed from the model's next-generation matrix. Thus, we quantified both the impact of age-stratified seroprevalence (assumed to be protective) on $R_{\text{eff}}$ as well as uncertainty in $R_{\text{eff}}$.

## Serosurvey sample size and allocation for inference and modeling

The flexible framework described in *Figure 1* enables the calculation of sample sizes for different serological survey designs. To calculate the number of tests required to achieve a seroprevalence estimate with a specified tolerance for uncertainty, and to determine optimal test allocation across subpopulations in the context of studying a particular intervention, we treated the estimate uncertainty as a framework output and then sought to minimize it by improving the allocation of samples (*Figure 1*, dashed arrow).

Uniform allocation of samples to subpopulations is not always optimal. It can be improved by (i) increasing sampling in subpopulations with higher seroprevalence and (ii) sampling in subpopulations with higher relative influence on the quantity to be estimated. This approach, which we term model and demographics informed (MDI), allocates samples to subpopulations in proportion to how much sampling them would decrease the posterior variance of estimates, that is, $n_i \propto x_i \sqrt{\theta_i^*(1 - \theta_i^*)}$, where $\theta_i^* = 1 - \text{sp} + \theta_i(\text{se} + \text{sp} - 1)$ is the probability of a positive test in subpopulation $i$ given test sensitivity (se), test specificity (sp), and subpopulation seroprevalence $\theta_i$, and $x_i$ is the relative importance of subpopulation $i$ to the quantity to be estimated.

The sample allocation recommended by MDI varies depending on the information available and the quantity of interest. When the key quantity is overall seroprevalence, $x_i$ is the fraction of the population in subpopulation $i$. When the key quantity is total infections, the effective reproductive number, $R_{\text{eff}}$, or another quantity derived from compartmental models with subpopulations, $x_i$, is the

*i*th entry of the principal eigenvector of the model's next-generation matrix, after modification to include modeled interventions. In such scenarios, this approach balances the importance of sampling subpopulations due to their role in dynamics ($x_i$) and higher variance in seroprevalence estimates themselves ($\sqrt{\theta_i^*(1 - \theta_i^*)}$). If subpopulation prevalence estimates $\theta_i$ are unknown, sample allocation based solely on $x_i$ is recommended. These methods are derived in Appendix A3.

## Data sources

Age distribution of U.S. blood donors was drawn from a study of Atlanta donors (*Shaz et al., 2011*). Age distribution of U.S. mothers was drawn from the 2016 CDC Vital Statistics Report using Massachusetts as a reference state (*Martin et al., 2018*). Daily age-structured contact data were drawn from *Prem et al., 2017*. All data were represented using 5-year age bins, that is, $(0 - 4, 5 - 9, \ldots, 74 - 79)$. For datasets with bins wider than 5 years, counts were distributed evenly into the 5-year bins. Serological test characteristics were collected from registrations with the *U.S. Food and Drug Administration, 2021* and summarized in *Supplementary file 1*. No attempt was made to test or validate manufacturer claims, and point estimates of sensitivity and specificity were used that did not incorporate test calibration sample sizes (*Gelman and Carpenter, 2020*; *Larremore and Fosdick, 2020*). Demographic data for the U.S., India, and Switzerland (analyzed in the article) as well as other countries (provided in open-source code) were downloaded from the 2019 United Nations World Populations Prospects report (*United Nations, 2019*). Hypothetical survey samples were drawn based on comprehensive seroprevalence estimates from Geneva, Switzerland (*Stringhini et al., 2020*).

## Results

### Test sensitivity/specificity, sampling bias, and true seroprevalence influence the accuracy and robustness of estimates

We simulated serological data from a single population with seroprevalence rates ranging from 1% to 50% using the reported sensitivity (90%) and specificity (>99.9%) of the Euroimmun SARS-CoV-2 IgG test (*U.S. Food and Drug Administration, 2021*; *Supplementary file 1*), and with the number of samples ranging from 100 to 5000. We constructed Bayesian posterior estimates of seroprevalence, finding that, when seroprevalence is 10% or lower, around 1000 samples are necessary to correctly estimate seroprevalence to within ±2% (*Figure 2*). Marketed tests with other characteristics also required around 1000 tests (*Figure 2—figure supplement 1A, B*) to achieve the same uncertainty levels, approaching the minimum sample size achieved by a theoretical test with perfect sensitivity and specificity (*Figure 2—figure supplement 1C*). Similar calculations for other test characteristics may be performed using the open-source tools that accompany this study (*Open-source code repository and reproducible notebooks for this manuscript, 2020*). In general, estimates were most uncertain when true seropositivity was near 50%, the number of samples was low, and/or test sensitivity/specificity were low.

Next, we tested the ability of the Bayesian hierarchical model to infer both population and subpopulation seroprevalence. We simulated serological data from subpopulations for which samples were allocated and with heterogeneous seroprevalence levels (*Supplementary file 2*) and average seroprevalence values between 5% and 50%. Test outcomes were randomly generated conditioning on the false positive and negative properties of the test being modeled (*Supplementary file 1*). Test allocations across subpopulations were specified in proportion to age demographics of blood donations, delivering mothers, uniformly across subpopulations, or according to an MDI allocation focused on minimized posterior uncertainty in $R_{\text{eff}}$.

Credible intervals of the resulting overall seroprevalence estimates were influenced by the age demographics sampled, with the most uncertainty in the newborn dried blood spots sample set, due to the narrow age range for the mothers (*Figure 3*). For such sampling strategies, which draw from only a subset of the population, our approach assumes that seroprevalence in each subpopulation does not dramatically vary and thus infers that seroprevalence in the unsampled bins is similar to that in the sampled bins but with increased uncertainty. Uncertainty was also influenced by the overall seroprevalence, such that the width of the 95% credible interval increased with higher seroprevalence for a given sample size. While test sensitivity and specificity also impacted uncertainty,

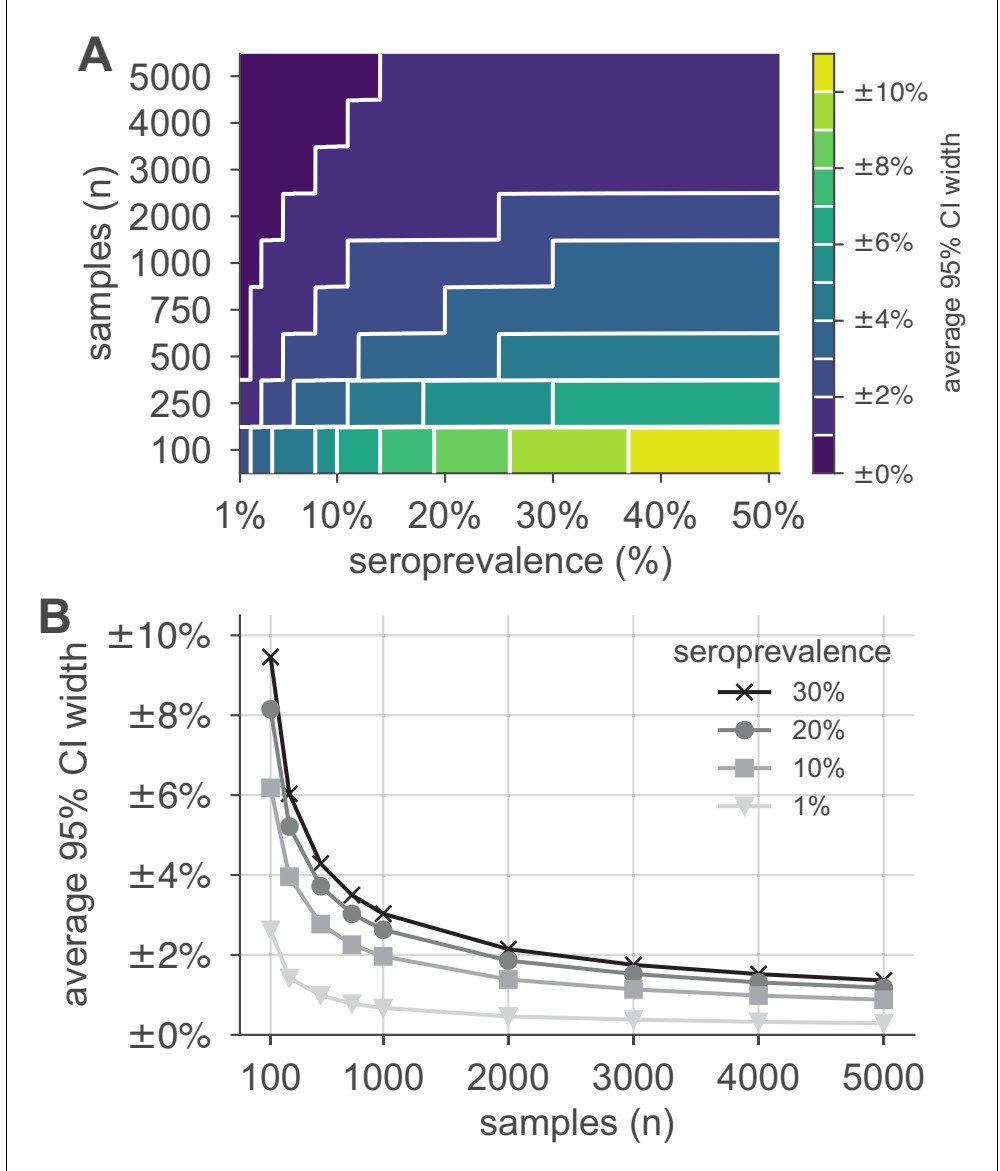

**Figure 2.** Uncertainty of population seroprevalence estimates as a function of number of samples and true population rate. Uncertainty, represented by the width of 95% credible intervals, is presented as ± seroprevalence percentage points in (A) a contour plot and (B) for selected seroprevalence values, based on a serological test with 90% sensitivity and >99.9% specificity (*Figure 2—figure supplement 1* depicts results for other sensitivity and specificity values). In total, 5000 samples are sufficient to estimate any seroprevalence to within a worst-case tolerance of ±1.4 percentage points (e.g., 20% ± 1.4% = [18.6%, 21.4%]), even with the imperfect test studied. Each point or pixel is averaged over 250 stochastic draws from the specified seroprevalence with the indicated sensitivity and specificity.

The online version of this article includes the following figure supplement(s) for figure 2:

**Figure supplement 1.** Uncertainty of population seroprevalence estimates as a function of number of samples and true population rate.

central estimates of overall seropositivity were robust for sampling strategies that spanned the entire population. Note that the MDI sample allocation shown in *Figure 3* was optimized to estimate the effective reproductive number $R_{eff}$, and thus, while it performs well, it is slightly outperformed by uniform sampling when used to estimate overall seroprevalence.

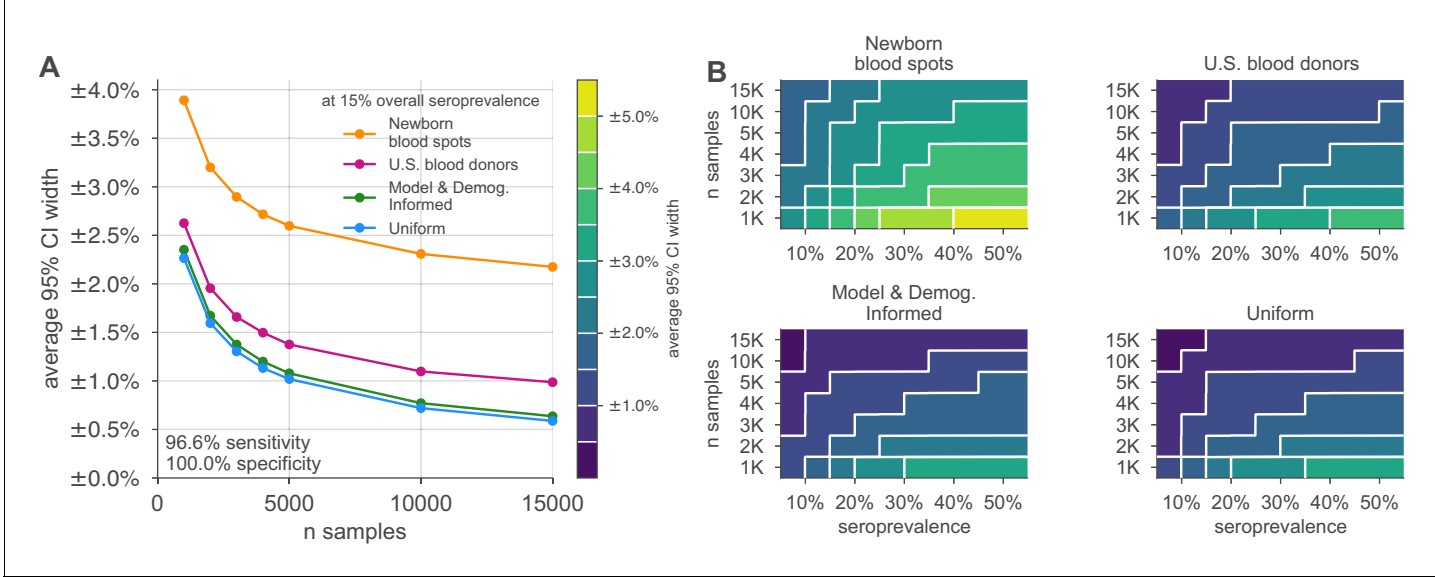

**Figure 3.** Uncertainty of overall seroprevalence estimates from convenience and formal sampling strategies. Uncertainty, represented by the width of 95% credible intervals, is presented as ± seroprevalence percentage points, based on a serological test with 90% sensitivity and >99.9% specificity (*Figure 3—figure supplement 1* depicts results for other sensitivity and specificity values). (A) Curves show the decrease in average CI widths for 15% seroprevalence, illustrating the advantages of using uniform and model and demographics informed (MDI) samples over convenience samples. (B) Contour plots show average CI widths for various total sample counts and overall seroprevalence ranging from 5% to 50%. Convenience samples derived from newborn blood spots (reflecting the age demographics of mothers) or U.S. blood donors improve with additional sampling but retain baseline uncertainty due to demographics not covered by the convenience sample. For the estimation of overall seroprevalence, uniform sampling is marginally superior to this example of the MDI sampling strategy, which was designed to optimize estimation of the effective reproductive number $R_{eff}$. Each point or pixel is averaged over 250 stochastic draws from the specified seroprevalence with the indicated sensitivity and specificity.

The online version of this article includes the following figure supplement(s) for figure 3:

**Figure supplement 1.** Uncertainty of overall seroprevalence estimates from convenience and formal sampling strategies.

## Seroprevalence estimates inform uncertainty in epidemic peak, timing, and reproductive number

*Figure 4* illustrates how the height and timing of peak infections varied in forward simulations under two serological sampling scenarios and two hypothetical social distancing policies for a basic SEIR framework parameterized using seroprevalence data. Uncertainty in seroprevalence estimates propagated through SEIR model outputs in stages: larger sample sizes at a given seroprevalence resulted in a smaller credible interval for the seroprevalence estimate, which improved the precision of estimates of both the height and timing of the epidemic peak. We note that seroprevalence estimates without correction for the sensitivity and specificity of the test resulted in biased estimates in spite of increasing precision with larger sample size (*Figure 4C, D*). Test characteristics also impacted model estimates, with more specific and sensitive tests leading to more precise estimates (*Figure 4— figure supplement 1*). Even estimations from a perfect test carried uncertainty corresponding to the size of the sample set (*Figure 4—figure supplement 1*).

*Figure 5* illustrates how the Bayesian hierarchical model extrapolates seroprevalence values in sampled subpopulations, based on convenience samples from particular age groups or age-stratified serological surveys, to the overall population, with uncertainty propagated from these estimates to model-inferred epidemiological parameters of interest, such as the effective reproduction number $R_{eff}$. Estimates from 1000 neonatal heel sticks or blood donations achieved more uncertain, but still reasonable, estimates of overall seroprevalence and $R_{eff}$ as compared to uniform or demographically informed sample sets (*Figure 5*). Here, convenience samples produced higher confidence estimates in the heavily sampled subpopulations, but high uncertainty estimates in unsampled populations through our Bayesian modeling framework. In all scenarios, our framework propagated uncertainty appropriately from serological inputs to estimates of overall seroprevalence (*Figure 5I*) or $R_{eff}$ (*Figure 5J*). Importantly, we note that the inferred posterior estimates shown in *Figure 5* are derived

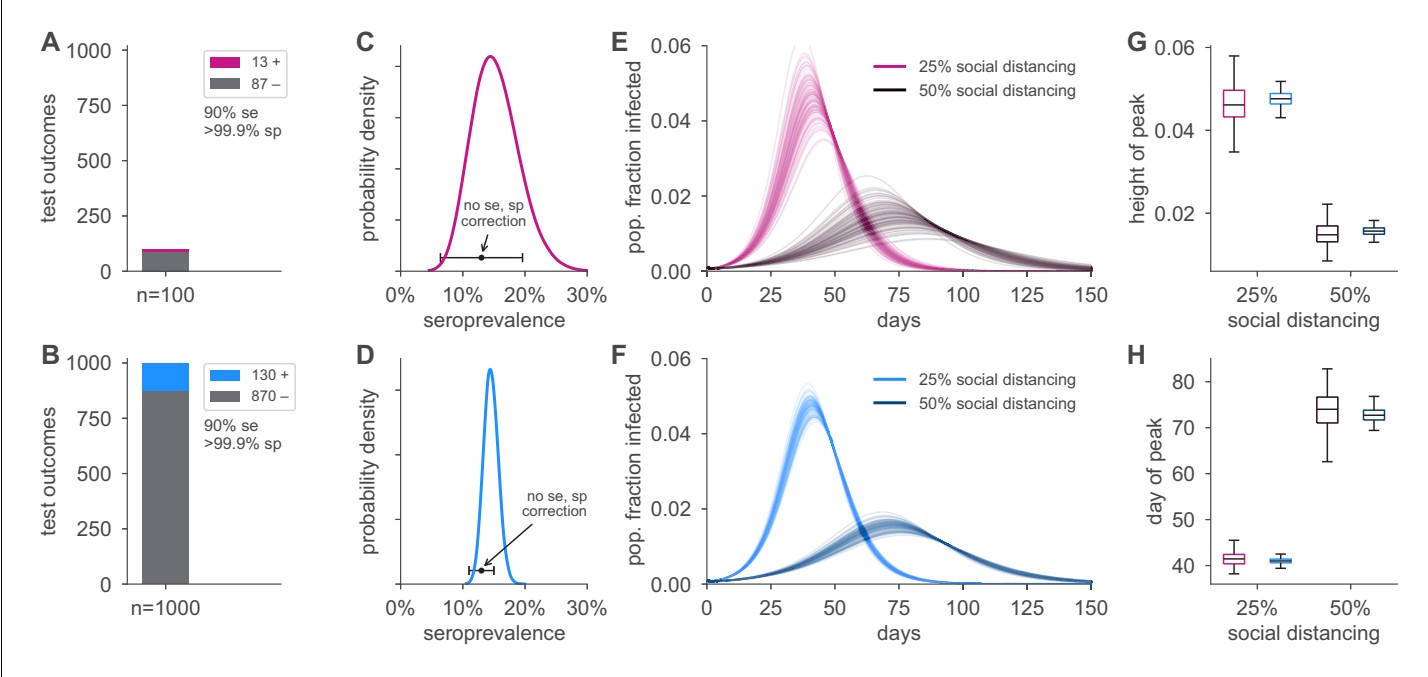

**Figure 4.** Uncertainty in serological data produces uncertainty in simulated epidemic peak height and timing. Serological test outcomes for $n = 100$ tests (A; red) and $n = 1000$ tests (B; blue) produce (C, D) posterior seroprevalence estimates with quantified uncertainty with posterior means of 15.2% and 14.6%, respectively; estimates uncorrected for assay performance bias: 13.0% and 13.0%. (E, F) Samples from the seroprevalence posterior produce a distribution of simulated epidemic curves for scenarios of 25% and 50% social distancing (see Materials and methods), leading to uncertainty in (G) epidemic peak and (H) timing, which is mitigated in the $n = 1000$ sample scenario. Boxplot whiskers span 1.5× IQR, boxes span central quartile, lines indicate medians, and outliers were suppressed. se, sensitivity; sp, specificity.

The online version of this article includes the following figure supplement(s) for figure 4:

**Figure supplement 1.** Uncertainty in serological data produces uncertainty in estimates of epidemic peak height and timing, even when the test has perfect sensitivity and specificity.

from stochastically generated data, meaning that repeating this numerical experiment would produce different simulated test outcomes and therefore different inferred seroprevalence and $R_{eff}$ estimates whose accuracy will stochastically vary, as expected. Improved test sensitivity and specificity correspondingly improved estimation and reduced the number of samples required (i) to achieve the same credible interval for a given seroprevalence and (ii) estimates of $R_{eff}$ (*Figure 5—figure supplements 1* and *2*).

If the subpopulations in the convenience sample have systematically different seroprevalence rates from the general population, increasing the sample size may bias estimates (*Figure 5—figure supplements 3* and *4*) while simultaneously decreasing the widths of posterior credible intervals, producing higher confidence in estimates in spite of their bias. This may be avoided using data from other sources or by updating the prior distributions in the Bayesian model with known or hypothesized relationships between seroprevalence of the sampled and unsampled populations. In general, the magnitude of this type of bias is not possible to estimate without secondary sources of seroprevalence data, differentiating it from the avoidable biases that result from failing to post-stratify based on population demographics or adjust for the sensitivity and specificity of the test instrument.

## Strategic sample allocation improves estimates

We used the MDI strategy to design a study that optimizes estimation of $R_{eff}$ and then tested the performance of the sample allocations against those resulting from blood donation and neonatal heel stick convenience sampling, as well as uniform sampling. As designed, MDI produced higher confidence posterior estimates (*Figure 5J*, *Figure 5—figure supplement 2*). Importantly, because the relative importance of subpopulations in a model varies based on the hypothetical interventions

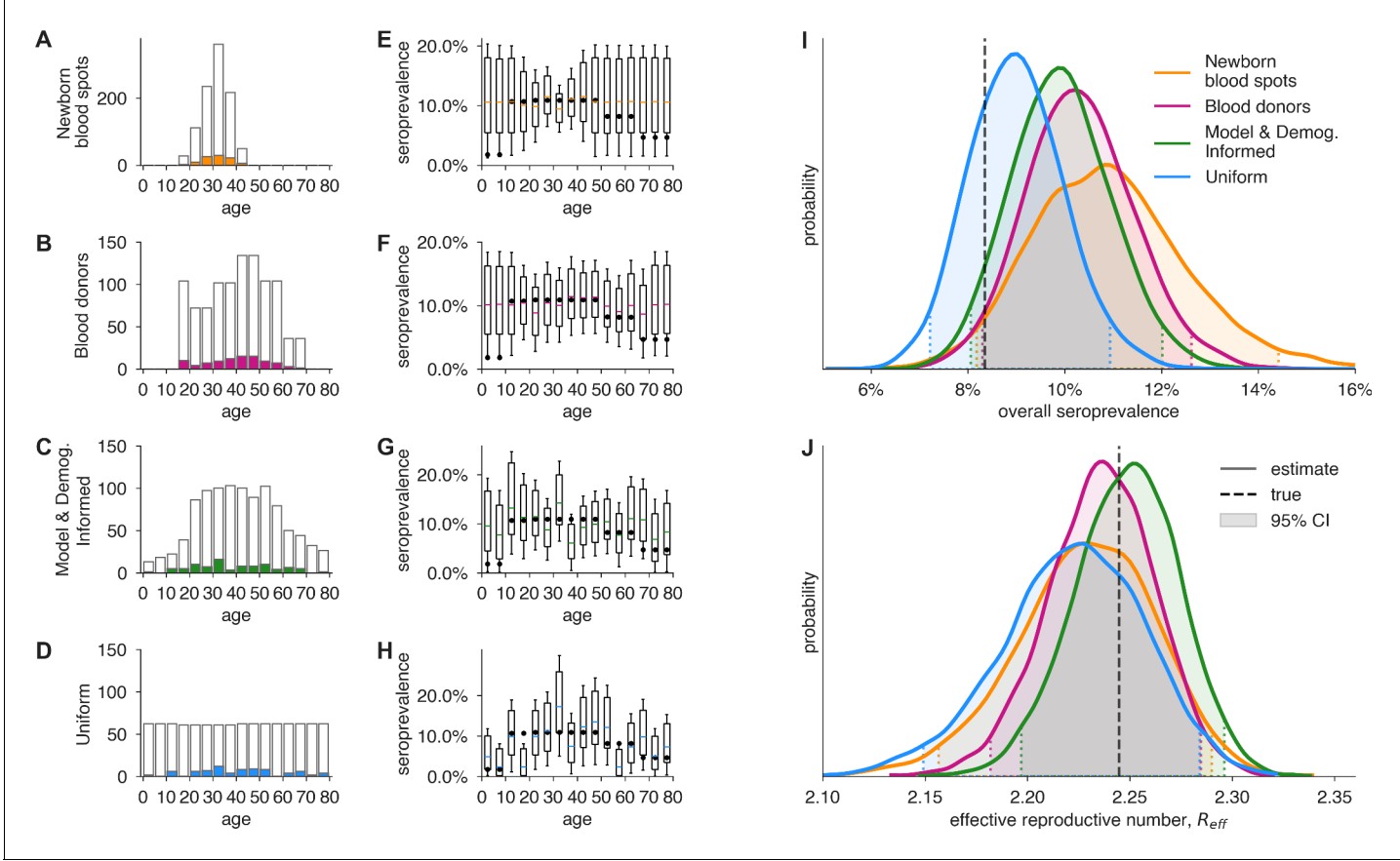

**Figure 5.** Convenience and formal samples provide serological and epidemiological parameter estimates. (**A–D**) For four sampling strategies, $n = 1000$ tests were allocated to age groups with negative tests (gray outlines) and positive tests (colors) as shown, drawn stochastically based on seroprevalence estimates reflecting SARS-CoV-2 serosurvey outcomes from Geneva, Switzerland, as of May 2020 (**Stringhini et al., 2020**) for a test with 90% sensitivity and >99.9% specificity. The model and demographics informed (MDI) strategy shown was designed to optimize estimation of $R_{eff}$. (**E–H**) Age-group seroprevalence estimates $\theta_i$ are shown as boxplots (boxes 90% CIs, whiskers 95% CIs); dots indicate the true values from which data were sampled (**Stringhini et al., 2020**). Note the decreased uncertainty for boxes with higher sampling rates. (**I**) Age-group seroprevalences were weighted by Swiss population demographics to produce overall seroprevalence estimates, shown as probability densities with 95% credible intervals shaded and highlighted with dashed lines. (**J**) Age-group seroprevalences were used to estimate the effective reproductive number ($R_{eff}$) from an age-stratified transmission model under *status quo ante* contact patterns, shown as probability densities with 95% credible intervals shaded and highlighted with dashed lines, based on a basic reproductive number in the absence of population immunity ($R_0$) of 2.5. Dashed lines indicate true values from which the data were sampled. Each distribution depicts inference outcomes from a single set of stochastically sampled data; no averaging is done. Note that although uniform and MDI sample allocation produces equivalently confident estimates of overall seroprevalence, MDI produces a more confident estimate of $R_{eff}$ because it allocates more samples to age groups most relevant to model dynamics.

The online version of this article includes the following figure supplement(s) for figure 5:

**Figure supplement 1.** Average credible interval width for overall seroprevalence estimates using four sampling strategies and four serological test kits.
**Figure supplement 2.** Average credible interval width for $R_{eff}$ estimates using four sampling strategies and four serological test kits.
**Figure supplement 3.** Credible interval coverage for overall seroprevalence estimates using four sampling strategies and four serological test kits.
**Figure supplement 4.** Credible interval coverage for $R_{eff}$ estimates using four sampling strategies and four serological test kits.

being modeled (e.g., the reopening of workplaces would place higher importance on the serological status of working-age adults), MDI sample allocation recommendations should be derived for multiple hypothetical interventions and then averaged to design a study from which the largest variety of high confidence results can be derived. To illustrate how such recommendations would work in practice, we computed MDI recommendations to optimize three scenarios for the contact patterns and demography of the U.S. and India, deriving a balanced sampling recommendation (**Figure 6**).

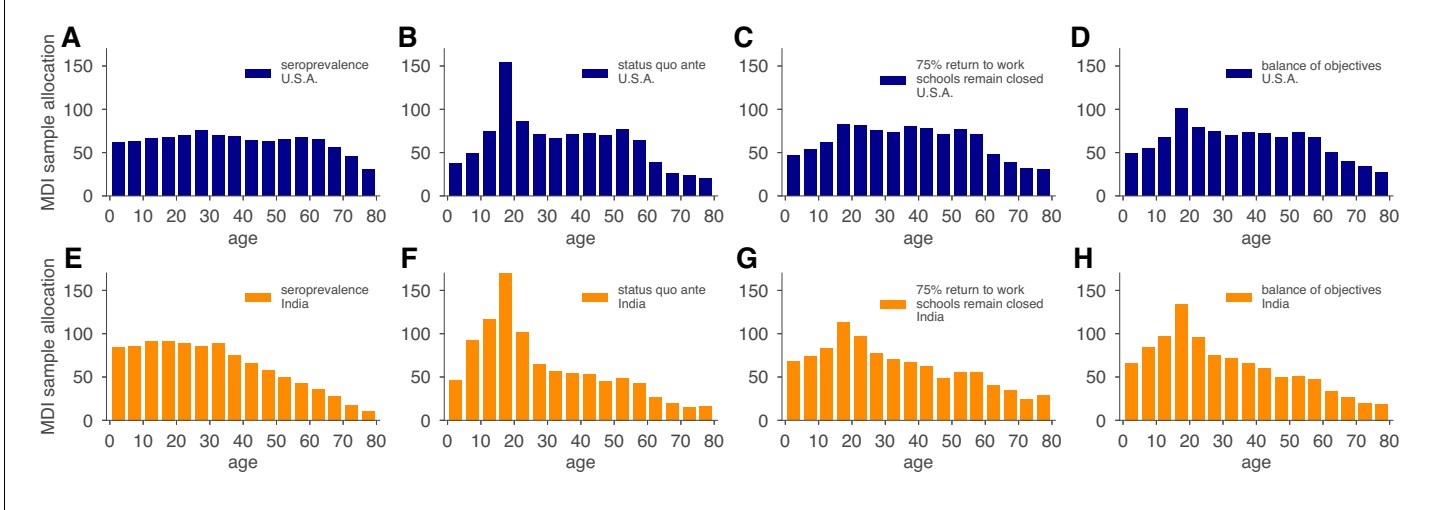

**Figure 6.** Model and demographics informed (MDI) sample allocations vary by demographics and modeling needs. Bar charts depict recommended sample allocation for three objectives, reducing posterior uncertainty for (A, E) estimates of overall seroprevalence, (B, F) predictions from an age-structured model with *status quo ante* contact patterns, (C, G) predictions from an age-structured model with modified contacts representing, relative to pre-crisis levels, a 20% increase in home contact rates, closed schools, a 25% decrease in work contacts, and a 50% decrease of other contacts (*Mossong et al., 2008*; *Prem et al., 2017*), and (D, H) averaging the other three MDI recommendations to balance competing objectives. Data for both the U.S. (blue; A–D) and India (orange; E–H) illustrate the impact of demography and contact structure on strategic sample allocation. These sample allocation strategies assume no prior knowledge of subpopulation seroprevalences $\{\theta_i\}$.

## Discussion

There is a critical need for serological surveillance of SARS-CoV-2 to estimate cumulative incidence. Here, we presented a formal framework for doing so to aid in the design and interpretation of serological studies, which avoids the biases associated with seroprevalence estimates that fail to account for sensitivity, specificity, and sampling schemes. We considered that sampling may be done in multiple ways, including efforts to approximate seroprevalence using convenience samples, as well as more complex and resource-intensive structured sampling schemes, and that these efforts may use one of any number of serological tests with distinct test characteristics. We incorporated into this framework an approach to propagating the estimates and associated uncertainty through mathematical models of disease transmission (focusing on scenarios where seroprevalence maps to immunity) to provide decision-makers with tools to evaluate the potential impact of interventions and thus guide policy development and implementation.

Our results suggest approaches to serological surveillance that can be adapted as needed based on pre-existing knowledge of disease prevalence and trajectory, availability of convenience samples, and the extent of resources that can be put towards structured survey design and implementation. While this work focuses on the design and analysis of single cross-sectional surveys, stratified by age, extensions to the analysis of serial cross-sectional surveys (*Stringhini et al., 2020*; *Nisar et al., 2020*) or other stratifications are also possible. Our results suggest that such surveys could benefit from the rebalancing of limited test budgets between subpopulations from one cross-sectional wave to the next by basing each wave's test allocation strategy on the MDI recommendations derived from the preceding wave. Although our numerical demonstrations here focused on heterogeneity and modeling by age, our work may be applied to any population stratification for which there is heterogeneity. Indeed, seroprevalence studies by neighborhood in New York City (*Nyc health testing data, 2020*), Karachi (*Nisar et al., 2020*), and Mumbai (*Malani et al., 2020*) have all found geographical variation in seroprevalence. In situations where sample sizes are low and heterogeneity is high, hyperprior parameters can be adjusted to accommodate larger variation between subpopulations.

In the absence of baseline estimates of cumulative incidence, an initial serosurvey can provide a preliminary estimate (*Figure 2*). Our framework updates the 'rule of 3' approach (*Hanley and*

*Lippman-Hand, 1983*) by incorporating uncertainty in test characteristics and can further address uncertainty from biased sampling schemes (see Appendix A4). As a result, convenience samples, such as the maternal antibodies within newborn heel stick dried blood spots or samples from blood donors, can be used to estimate population seroprevalence. However, it is important to note that in the absence of reliable assessment of correlations in seroprevalence across age groups, extrapolations from these convenience samples to entire populations may be misleading as sample size increases (*Figure 5—figure supplement 3*). Indeed, as convenience sample size increases, credible intervals will shrink, which, if sampled groups are unrepresentative of unsampled groups, will constitute a 'false precision'. Uniform or model and demographic informed samples, while more challenging logistically to implement, give the most reliable estimates. The results of a one-time study could be used to update the priors of our Bayesian hierarchical model and improve the inferences from convenience samples. In this context, we note that our framework naturally allows the integration of samples from multiple test kits and protocols, provided that their sensitivities and specificities can be estimated (*Larremore and Fosdick, 2020*; *Gelman and Carpenter, 2020*), which will become useful as serological assays improve in their specifications.

The results from serological surveys will be invaluable in projecting epidemic trajectories and understanding the impact of interventions such as age-prioritized vaccination (*Bubar et al., 2021*). We have shown how the estimates from these serological surveys can be propagated into transmission models, incorporating model uncertainty as well. Conversely, to aid in rigorous assessment of particular interventions that meet accuracy and precision specifications, this framework can be used to determine the needed number and distribution of population samples via model and demographic-informed sampling. Extensions could conceivably address other study planning questions, including sampling frequency (*Herzog et al., 2017*).

There are a number of limitations to this approach that reflect uncertainties in the underlying assumptions of serological responses and the changes in mobility and interactions due to public health efforts (*Kissler et al., 2020b*). Serology reflects past infection, and the delay between infection and detectable immune response means that serological tests reflect a historical cumulative incidence (the date of sampling minus the delay between infection and detectable response). However, due to the waning of antibody concentrations over time (*Ward et al., 2020*), seroreversion may cause seroprevalence studies to underestimate cumulative incidence. As a consequence, modeling studies that incorporate seroprevalence estimates should acknowledge such potential delays and seroreversion when interpreting their findings. The possibility of heterogeneous immune responses to infection and unknown dynamics and duration of immune response means that interpretation of serological survey results may not accurately capture cumulative incidence. For COVID-19, we do not yet understand the serological correlates of protection from infection, and as such projecting seroprevalence into models that assume seropositivity indicates immunity to reinfection may be an overestimate; models would need to be updated to include partial protection or return to susceptibility.

Our work also requires the specification of prior and hyperprior distributions, assumptions inherent to any Bayesian approach to statistical inference. Here, we used uninformative uniform prior distributions and a weakly informative hyperprior distribution in order to impose minimal assumptions when modeling the data. This is a conservative choice as assuming uninformative prior distributions results in higher posterior uncertainty. While informative priors can reduce uncertainty in seroprevalence studies (*Gelman and Carpenter, 2020*), specifying such priors appropriately relies on additional information and/or assumptions about the study population, which may be sparse, particularly during an unfolding pandemic.

Use of model and demographic-informed sampling schemes is valuable for projections that evaluate interventions but are dependent on accurate parameterization. While in our examples we used POLYMOD and other contact matrices, these represent the *status quo ante* and should be updated to the extent possible using other data, such as those obtainable from surveys (*Mossong et al., 2008*; *Prem et al., 2017*) and mobility data from online platforms and mobile phones (*Buckee et al., 2020*; *Ainslie et al., 2020*; *Open COVID-19 Data Working Group et al., 2020*). Moreover, the framework could be extended to geographic heterogeneity as well as longitudinal sampling if, for example, one wanted to compare whether the estimated quantities of interest (e.g., seroprevalence, $R_{\mathrm{eff}}$) differ across locations or time (*Abrams et al., 2014*; *Stringhini et al., 2020*; *Kissler et al., 2020a*).

Here, we explored only SEIR models, but extensions to alternatives that incorporate waning immunity (*Ward et al., 2020*) and a return to full or partial susceptibility are possible (*Saad-Roy et al., 2020*). Clarified understanding of SARS-CoV-2 antibody titers, protection, and durability will further inform whether it is appropriate to model seropositive individuals as no longer susceptible, as we did in example calculations here. We note that, across model types, the derivation of model-focused MDI sample allocation strategies requires only the formulation of a next-generation matrix or network of subpopulations' epidemiological impacts on each other, providing a more general framework spanning model assumptions and classes.

Overall, the framework here can be adapted to communities of varying size and resources seeking to monitor and respond to SARS-CoV-2 and future pandemics. Further, while the analyses and discussion focused on addressing urgent needs, this is a generalizable framework that with appropriate modifications can be applicable to other infectious disease epidemics.

## Acknowledgements

The authors thank Nicholas Davies, Laurent Hébert-Dufresne, Johan Ugander, Arjun Seshadri, and the BioFrontiers Institute IT HPC group. The work was supported in part by the Morris-Singer Fund for the Center for Communicable Disease Dynamics at the Harvard T.H. Chan School of Public Health. DBL and YHG were supported in part by the SeroNet program of the National Cancer Institute (1U01CA261277-01). Reproduction code is open source and provided by the authors at github.com/LarremoreLab/covid_serological_sampling (*Larremore, 2021*; copy archived at swh:1:rev:262fb34c19c4bb48bdc74dad1470e4bf8bbe5a69).

## Additional information

### Funding

| Funder | Grant reference number | Author |
|---|---|---|
| Morris-Singer Fund for the Center for Communicable Disease Dynamics | | Stephen M Kissler<br>Caroline O Buckee<br>Yonatan H Grad |
| National Cancer Institute | 1U01CA261277-01 | Daniel B Larremore<br>Yonatan H Grad |

The funders had no role in study design, data collection and interpretation, or the decision to submit the work for publication.

### Author contributions

Daniel B Larremore, Conceptualization, Data curation, Software, Formal analysis, Supervision, Investigation, Visualization, Methodology, Writing - original draft, Project administration, Writing - review and editing; Bailey K Fosdick, Conceptualization, Formal analysis, Investigation, Writing - original draft, Writing - review and editing; Kate M Bubar, Data curation, Formal analysis, Writing - original draft, Writing - review and editing; Sam Zhang, Software, Visualization, Writing - original draft, Writing - review and editing; Stephen M Kissler, Formal analysis, Investigation, Methodology, Writing - original draft, Writing - review and editing; C Jessica E Metcalf, Caroline O Buckee, Conceptualization, Writing - original draft, Writing - review and editing; Yonatan H Grad, Conceptualization, Formal analysis, Supervision, Investigation, Methodology, Writing - original draft, Project administration, Writing - review and editing

### Author ORCIDs

Daniel B Larremore (iD) https://orcid.org/0000-0001-5273-5234
Stephen M Kissler (iD) http://orcid.org/0000-0003-3062-7800
C Jessica E Metcalf (iD) http://orcid.org/0000-0003-3166-7521
Caroline O Buckee (iD) https://orcid.org/0000-0002-8386-5899
Yonatan H Grad (iD) https://orcid.org/0000-0001-5646-1314

Decision letter and Author response
Decision letter https://doi.org/10.7554/eLife.64206.sa1
Author response https://doi.org/10.7554/eLife.64206.sa2

## Additional files

### Supplementary files

• Supplementary file 1. Serological tests and performance characteristics considered in this study. Sensitivity and specificity values were taken from manufacturer's claims as of February 2021 as filed with the *U.S. Food and Drug Administration, 2021*.

• Supplementary file 2. Parameter values used in dynamical models and numerical experiments. This table is divided into four sections. The top two sections correspond to the parameters of the single-population modeling. The bottom two sections correspond to the parameters used in the age-structured modeling. Sections corresponding to $\theta$ and $\theta_i$ are separated to indicate that their values were used in synthetic data experiments to assess performance of the Bayesian inference methods on varying seroprevalence levels, to separate them from SEIR model parameters. Contact matrices $C_{ij}$ used in this article were, in particular, those corresponding to the U.S., India, and Switzerland. Equations for models can be found in the Appendix. Test kit sensitivity and specificity values are provided in *Supplementary file 1*. Subpopulation seropositivity values $\theta_i$ were synthetically generated to accommodate moderate variation between subpopulations as well as the ability for mean seropositivity to be easily adjusted (used in *Figure 3*, or were generalized to 5-year age bins from the age-stratified serosurvey of *Stringhini et al., 2020*) based in Geneva, Switzerland, as shown (used in *Figure 5*).

• Transparent reporting form

### Data availability

Reproduction code is open source and provided by the authors at http://github.com/LarremoreLab/covid_serological_sampling (copy archived at https://archive.softwareheritage.org/swh:1:rev:262fb34c19c4bb48bdc74dad1470e4bf8bbe5a69/).

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

## Appendix 1

### A1 Bayesian inference of seroprevalence

#### A1.1 Inference of seroprevalence in a sample using an imperfect test

If a serological test had perfect sensitivity and specificity, the probability of observing $n_+$ seropositive results from $n$ tests, given a true population seroprevalence $\theta$ is given by the binomial distribution:

$$\Pr(n_+ \mid \theta) = \binom{n}{n_+} \theta^{n_+} (1-\theta)^{n-n_+}, \qquad \theta \in [0,1]. \tag{A1}$$

However, imperfect specificity and sensitivity require that we modify this formula. For convenience, in the remainder of this Appendix, we will use

$$u \equiv \Pr(\text{test is positive} \mid \text{seronegative}) = 1 - \text{specificity}$$
$$v \equiv \Pr(\text{test is negative} \mid \text{seropositive}) = 1 - \text{sensitivity}$$

Using this notation, the probability that a single test returns a positive result, given $u$, $v$, and the true seroprevalence $\theta$, is

$$\Pr(\text{test is positive} \mid \theta, u, v) = \theta(1-v) + (1-\theta)u. \tag{A2}$$

Substituting this per-sample probability into *Equation (A1)* yields

$$\Pr(n_+ \mid \theta, u, v) = \binom{n}{n_+} [u + \theta(1-u-v)]^{n_+} [1 - u - \theta(1-u-v)]^{n-n_+}. \tag{A3}$$

Note that *Equation (A1)*, and therefore *Equation (A3)*, both assume that samples are drawn independently and can therefore be computed using a binomial likelihood. This assumption may be modified in scenarios in which factors contributing to non-independence of samples are known and measured, for example, when individuals are sampled but are known to belong to the same household (*Nisar et al., 2020*). Finally, using Bayes' rule, we can write the posterior distribution over seropositivity $\theta$, given the data, the test's parameters (*Diggle, 2011*), and an uninformative (uniform) prior on $\theta$, yielding

$$\Pr(\theta \mid n_+, u, v) = \frac{[u + \theta(1-u-v)]^{n_+} \, [1 - u - \theta(1-u-v)]^{n-n_+}}{\left[ \frac{B(1-v,1+n_+,1+n-n_+) - B(u,1+n_+,1+n-n_+)}{1-u-v} \right]}, \tag{A4}$$

where $B$ is an *incomplete beta function* without normalization. In practice, to sample from this distribution, one can use an accept–reject algorithm with, for example, a uniform proposal distribution and consider only the numerator of *Equation (A4)*. Alternatively, one can generate samples from a truncated beta distribution using accept–reject sampling or an inverse cumulative distribution function method, and these samples can be transformed to represent draws from *Equation (A4)*.

#### A1.2 Bayesian estimation of seroprevalence across subpopulations

For a test with sensitivity $1-v$ and specificity $1-u$, and given $n_{i+}$ seropositive results from $n_i$ tests in subpopulation $i$—set equal to zero for unsampled subpopulations—the posterior distribution over the vector of subpopulation seropositivities $\theta = \{\theta_i\}$ given all results $n_+ = \{n_{i+}\}$ is given by

$$\Pr(\theta \mid n_+, u, v) = \iint_{\bar{\theta}, \gamma} \Pr(\theta, \bar{\theta}, \gamma \mid n_+, u, v) \, d\bar{\theta} \, d\gamma \tag{A5}$$

where we have included a hierarchy of priors. Specifically, the prior for each subpopulation seroprevalence was $\theta_i \sim \text{Beta}(\bar{\theta}\gamma, (1-\bar{\theta})\gamma)$, which has expectation $\bar{\theta}$ and variance $\bar{\theta}(1-\bar{\theta})/(\gamma+1)$. The hyperprior for the overall mean $\bar{\theta}$ was uniform on the interval $(0,1)$, allowing it to be dictated by the observed data. The hyperprior for the variance parameter was $\gamma \sim \text{Gamma}(\nu, \text{scale} = \gamma_0/\nu)$, which has expected value $E[\gamma] = \gamma_0$ and $Var[\gamma] = \gamma_0^2/\nu$.

## A1.3 Sampling from the Bayesian hierarchical model for subpopulation seroprevalences using MCMC

We sample from the joint posterior distribution inside the integral in *Equation (A5)* using a MCMC algorithm, with univariate Metropolis–Hastings updates. We initialize the age-specific seroprevalence parameters at $\theta_i = (n_+ + 1)/(n_i + 2)$, set $\bar{\theta}$ equal to the sample mean of the $\{\theta_i\}$ and set $\gamma = \gamma_0$. For each simulation, the MCMC algorithm was run for a total of 50,100 iterations. The first 100 iterations were discarded and every 50th sample was saved to obtain 1000 samples from the joint posterior distribution. Code is open source and freely available (https://github.com/LarremoreLab/covidserologicalsampling). Trace plots and effective sample sizes of the posterior samples were used to evaluate convergence and mixing of the chains. Effective sample sizes were greater than 6000 for all parameters, and trace plots did not raise concerns about the MCMC algorithm settings.

## A2 Including protective seropositivity into models

### A2.1 Canonical single-population SEIR model with social distancing and seropositivity

Let $S$, $E$, $I$, and $R$ be the number of susceptible, exposed, infected, and recovered people in a population of size $N$, $S + E + I + R = N$. We model dynamics by

$$
\begin{aligned}
\dot{S} &= -\beta\rho SI \\
\dot{E} &= \beta\rho SI - \alpha E \\
\dot{I} &= \alpha E - \gamma I \\
\dot{R} &= \gamma I
\end{aligned}
\tag{A6}
$$

where $\beta$, $\alpha$, and $\gamma$ represent the rates of infection, symptom onset, and recovery, respectively, as in a standard SEIR model. To model social distancing, we include the contact parameter $\rho \in [0,1]$, which modulates the fraction of social contacts between $S$ and $I$ populations that remain. Thus, $\rho = 1$ represents no social distancing while $\rho = 0.5$ would represent a 50% reduction in contacts. In the simulations of this article, only $\rho = 0.5, 0.75$ were considered as examples of dynamics.

To parameterize this model using seroprevalence, we made the modeling assumption that seropositive individuals are immune. Noting that this is only an assumption that at present requires in-depth research, we therefore placed seropositive individuals into the recovered group. In other words, for a seropositive fraction $\theta$, with 10 individuals in the $E$ and $I$ compartments each, initial conditions would be

$$
(S_0, E_0, I_0, R_0) = (N - \theta N - 20, \ 10, \ 10, \ \theta N).
$$

### A2.2 Canonical single-population SEIR parameters and simulation details

Parameter values used in this study can be found in *Supplementary file 2*. In prose, the model used transmission rate $\beta = 1.75$, exposure-to-infected rate $\alpha = 0.2$, and recovery rate $\gamma = 0.5$, with no births or deaths, in a finite population of size $N = 10,000$. Social distancing was implemented as a coefficient $\rho = \{0.5, 0.75\}$, corresponding to 50% and 25% social distancing, multiplying the contact rate between infected and susceptible populations. Integration was performed for 150 days with a timestep of 0.1 days. Initial conditions for $(S, E, I, R)$ were $(N - 20 - \theta N, 10, 10, \theta N)$, to simulate a fraction $\theta$ of recovered individuals, assumed to be immune. For each sampled value of $\theta$, peak infection height and timing were extracted from forward-integrated time series.

### A2.3 Age-structured (POLYMOD) SEIR model with seropositivity

A model with 16 age bins $(0-4, 5-9, \ldots 75-79)$ was parameterized using country-specific age-contact patterns (*Mossong et al., 2008*; *Prem et al., 2017*) and COVID-19 parameter estimates (*Davies et al., 2020*). The model includes age-specific clinical fractions and varying durations of

preclinical, clinical, and subclinical infectiousness, as well as a decreased infectiousness for sub-clinical cases (*Davies et al., 2020*).

Davies et al. define a next-generation matrix

$$N_{ij} = u_i C_{ij} \left[ y_j (\mu_P + \mu_C) + (1 - y_j) f \mu_S \right] , \qquad (A7)$$

where $u_i$ is the susceptibility of age group $i$; $C_{ij}$ is the number of age-$j$ individuals contacted by an age-$i$ individual per day; $y_j$ is the probability that an infection is clinical for an age-$j$ individual; $\mu_P$, $\mu_C$, and $\mu_S$ are mean durations of preclinical, clinical, and subclinical infectiousness, respectively; and $f$ is the relative infectiousness of subclinical cases (*Davies et al., 2020*). Values for all parameters are reported in *Supplementary file 2*.

Protective seropositivity can be included in the model by multiplying $N_{ij}$ as defined above by $1 - \theta_i$, where $\theta_i$ is the seropositivity rate of age group $i$. With this included term, we can modify *Equation (A7)* as

$$\widetilde{N}(\boldsymbol{\theta}) = (I - D_{\boldsymbol{\theta}})N = (I - D_{\boldsymbol{\theta}})D_{\boldsymbol{u}}CD_{ay+b} , \qquad (A8)$$

where $D_x$ represents a diagonal matrix with entries $D_{ii} = x_i$, and the constants are defined $a = \mu_P + \mu_C - f\mu_S$ and $b = \mu_S$.

The effective reproductive number is then the spectral radius $\rho$ (i.e., the largest eigenvalue $\lambda$) of the next-generation matrix:

$$R_{\text{eff}}(\theta) = \rho\left[\widetilde{N}(\theta)\right] . \qquad (A9)$$

As written, *Equation (A9)* represents a model component shown in *Figure 1* (blue annotations) as it maps parameters $\theta$ to a point estimate of $R_{\text{eff}}$. As with the canonical SEIR model, uncertainty in the model parameters themselves can also be incorporated into overall uncertainty in $R_{\text{eff}}$ via Monte Carlo.

## A2.4 Age-structured SEIR model parameters and simulation details

Parameter values used in this study can be found in *Supplementary file 2* and were generally drawn from the work of Davies et al. and the sources therein. Published estimated contact matrices were used for India and the U.S. in the article, with additional countries' contact matrices shown in the accompanying open-source code.

## A3 MDI sampling

The calculations that follow rely on facts from optimization theory. We briefly review these here before applying these results in what follows.

Let $n = (n_1, ..., n_K)$. Suppose we want to minimize a function of the form

$$f(n) = \sum_i \frac{c_i}{n_i}, \qquad (A10)$$

subject to the constraint that $\sum_i n_i = n$. Using the method of Lagrange multipliers, it can be shown that $f(n)$ is minimized when $n_i \propto \sqrt{c_i}$. We apply this result below with various expressions for $c_i$ to determine the optimal allocation of $n$ tests across subpopulations in order to minimize the uncertainty of quantities of interest.

### A3.1 Minimizing posterior uncertainty for seroprevalence

Given age-specific seroprevalence estimates $\theta$, the estimate for overall seroprevalence is defined as $\theta_{pop} = \sum_i d_i \theta_i$, where $d_i$ is the proportion of the population in group $i$. The uncertainty of this estimator depends on the uncertainties of the age-specific seroprevalences, which inherently depend on the number of tests $n_i$ allotted to each subpopulation. Although the posterior uncertainties of the subpopulation seroprevalences are not available in closed form, we can nevertheless approximate them using the uncertainties in the corresponding maximum likelihood estimators. Here, we consider the maximum likelihood estimators based on a separate binomial model for each subpopulation,

that is, models of the form *Equation (A3),* where $\theta$ is replaced by $\theta_i$. Note that this model assumes independence among the subpopulation seroprevalences.

The maximum likelihood estimate of $\theta_i$, given $n_{i,+}$ positive tests out of $n_i$ tests administered, is

$$\hat{\theta}_i = \frac{n_{i,+}/n_i - u}{1 - u - v} \,,$$

but this is only valid when both the numerator and denominator are positive, corresponding to a value of $\hat{\theta}_i$ in the interval (0, 1). If the above estimator is computed and found to be negative, which happens when the fraction of tests that are positive is below the false positive rate, then the maximum likelihood lies at the end point, $\hat{\theta}_i = 0$. Similarly, if the estimator is found to be greater than one, $\hat{\theta}_i = 1$. These estimators are undefined if no tests are allocated to group $i$, that is, when $n_i = 0$.

Using the maximum likelihood estimators as proxies for the subpopulation posterior distributions, we can approximate the posterior variance of $\theta_{pop}$ as

$$\begin{aligned}\mathrm{Var}[\theta_{pop}] \quad &\approx \sum_i d_i^2 \mathrm{Var}[\hat{\theta}_i]\\&= \sum_i d_i^2 \frac{[u+\theta_i(1-u-v)][1-u-\theta_i(1-u-v)]}{n_i(1-u-v)^2},\end{aligned} \tag{A11}$$

where $\theta_i$ is the true seroprevalence of group $i$. This variance equation has the form of *Equation (A10),* and thus the optimal allocation of samples is given by $n_i$

$$n_i \propto d_i \sqrt{[u+\theta_i(1-u-v)][1-u-\theta_i(1-u-v)]}, \tag{A12}$$

where multiplicative constants have been absorbed into the proportion. In the absence of knowledge about the true subpopulation seroprevalences $\theta$, we recommend simply allocating samples with respect to the demographic information: $n_i \propto d_i$.

## A3.2 Minimizing posterior uncertainty for modeling

When the primary quantity of interest is the output from a model, improved test allocation strategies can be developed by leveraging the model structure. For example, suppose the goal is accurate estimation of the total number of infected individuals at some future time point $t$. To avoid confusion with the identity matrix $I$ or the subpopulation index $i$, let $h^t = (h_1^t, h_2^t, ...)$ denote the vector containing the number of infected individuals within each subpopulation and let the total number of infected individuals be $H^t = \sum_i h_i^t$. Using the next-generation matrix defined in *Equation (A7)* and modification for the depletion of susceptibles as in *Equation (A8)*, the next=generation matrix updates the vector of infected individuals per subpopulation as

$$\begin{aligned}\boldsymbol{h}^{t+1} \quad &= (I - D_\theta)N\boldsymbol{h}^t\\&\approx (I - D_\theta)k\lambda\boldsymbol{x}\end{aligned} \tag{A13}$$

where $x$ represents the normalized eigenvector of $N$ corresponding to the largest eigenvalue $\lambda$, and $k$ is a scalar $k = x^T h^t$. The next-generation matrix $N$ is non-negative and satisfies the conditions of the Perron–Frobenius theorem, which means that it has a largest eigenvalue $\lambda$—for a next-generation matrix, $R_0 = \lambda$—which is greater than or equal to all other eigenvalues, with a corresponding eigenvector $x$ of non-negative components. This means that repeated applications of $N$ to any initial vector that is not orthogonal to $x$ will become increasingly parallel to $x$ at a rate of $\lambda/|\lambda_2|$ per iteration, where $\lambda_2$ is the second largest eigenvalue of $N$. This is the basis of the so-called Power Method, which repeatedly applies the matrix to find the largest eigenvalue and its corresponding eigenvector. Rewriting *Equation (A13)* for each subpopulation $i$ leads to

$$\begin{aligned}h_i^{t+1} \quad &= (1 - \theta_i)\sum_j N_{ij}h_j^t\\&\approx (1 - \theta_i)(\boldsymbol{x}^T\boldsymbol{h}^t)\lambda x_i \,.\end{aligned} \tag{A14}$$

Note that as seroprevalence increases, $(1-\theta)$ approaches zero, thereby accounting for the

depletion of susceptibles in subpopulation $i$ by reducing the number of infected individuals therein in the next timestep.

There are two helpful interpretations of *Equations (A13) and (A14)*. First, the vector $x$ is the principal 'direction' of the next-generation matrix, and repeated iterations of the dynamics in a large population will result in infected fractions that are proportional to $x$. In the above, we approximate the effect of $N$ on $h$ as $k\lambda x$, an approximation that is better when $\lambda$ is well separated from the second eigenvalue $\lambda_2$. Measurements of $\lambda/|\lambda_2|$ for models considered in this article ranged from 2 to 4.

A second interpretation of this result appeals to the notion of the next-generation matrix $N$ as a *network* in which the nodes are infected subpopulations and the directed links $N_{ij}$ explain the effects of an infection at node $j$ on future infections at node $i$. In this network dynamical system, by calculating $x$ we have computed the *eigenvector centralities* of the network's nodes (*Mark Newman, 2018*), which are a measure of the importance of each subpopulation in the network.

With these preliminary calculations in mind, we turn to the estimation of $H^t$. Because $H^t = \sum_i h_i^t$, and because the values $h_i^t$ are all functions of a random variable $\theta$, $H^t$ is also a random variable. Our goal is to minimize its variance by strategically allocating finite samples in order to minimize the *important* posterior variances among the elements of $\theta$. In plain language, some of the subpopulations are more important in shaping future disease dynamics than others, so MDI will preferentially allocate more samples to those subpopulations in a principled manner, which we now derive.

As in *Equation (A11)*, we approximate the posterior variance of $\theta$ by the posterior variance of the corresponding maximum likelihood estimator $\hat{\theta}$. This results in the following approximation of the variance of the total number infected:

$$\begin{aligned}\text{Var}[H^t] &\approx \text{Var}\left[\sum_i (1-\hat{\theta}_i)\alpha_1\lambda_1 x_i\right]\\ &= \sum_i (\alpha_1\lambda_1 x_i)^2 \frac{[u+\theta_i(1-u-v)][1-u-\theta_i(1-u-v)]}{n_i(1-u-v)^2}.\end{aligned}$$

where $x_i$ is the $i$th element of the principal eigenvector $x$. The first expression is obtained by using the approximation in *Equations (A13) and (A14)*. The resulting variance expression has the form of *Equation (A10)*, and thus, ignoring constants, the optimal allocation of samples is given by $n_i$

$$\propto x_i \sqrt{[u+\theta_i(1-u-v)][1-u-\theta_i(1-u-v)]}. \tag{A15}$$

In the absence of knowledge about the true subpopulation seroprevalences $\theta$, we recommend simply allocating samples with respect to the entries of the principal eigenvector: $n_i \propto x_i$.

## A4 Impact of sensitivity and specificity on the 'rule of 3'

Suppose we have a perfect test ($u = v = 0$) and when we perform $n$ tests, zero are positive. The maximum likelihood estimate of the seroprevalence would be 0. *Hanley and Lippman-Hand, 1983* proposed a simple upper 95% confidence bound on true seroprevalence equal to $3/n$.

The derivation of this rule is motivated by the following question: 'What is the maximum seroprevalence under which the probability of observing zero positives in $n$ tests is less than or equal to 5%?'. Briefly, the probability of a negative test is $\theta$. and thus the probability of observing $n$ negative tests is $(1-\theta)^n$. Setting this equal to 0.05 and solving for $\theta$, we find $\theta = 1 - .05^{1/n} \approx 3/n$, where the approximation is based on the power series representation of the exponential function.

Now, let us consider what happens if sensitivity and specificity are not equal to one and again zero positive tests are observed. The probability of a negative test is then $1 - u - \theta(1-u-v)$. An upper 95% confidence bound on the true seroprevalence is then

$$\theta = \frac{1-u-.05^{1/n}}{1-u-v} \approx \frac{3/n-u}{1-u-v}, \tag{A16}$$

where the approximation is derived in a similar manner. Notice if $u > 3/n$, this upper bound is less than zero. This occurs when there is inconsistency between the specified false positive rate $u$ and the observed data; namely, this occurs when $n$ is large enough that we would have expected at least one false positive.

Even if seroprevalence is zero, we expect to observe some number positive tests simply due to imperfect test specificity. Suppose we observe $n_+$ positive tests from a sample of $n$. An approximate upper 95% confidence bound on the true seroprevalence:

$$\frac{\left(\frac{n_+}{n} - u\right) + 1.64\sqrt{\frac{n_+(n-n_+)}{n^3}}}{1 - u - v} \tag{A17}$$

