## [Decision Letter]

**Acceptance summary:**

The paper by Larremore et al. presents a Bayesian framework to incorporate several sources of uncertainty (test characteristics, sample size, and heterogeneity across tested subpopulations) into estimates of SARS-CoV-2 seroprevalence and derived epidemiological/transmission parameters. They then use this framework to optimize study design and sampling schemes for serosurveys. While none of the methods presented are novel per se, the paper does present a much needed formal framework to guide the analysis and design of SARS-CoV-2 serosurveys.

**Decision letter after peer review:**

Thank you for submitting your article "Estimating SARS-CoV-2 seroprevalence and epidemiological parameters with uncertainty from serological surveys" for consideration by *eLife*. Your article has been reviewed by two peer reviewers, and the evaluation has been overseen by a Reviewing Editor and Miles Davenport as the Senior Editor. The following individuals involved in review of your submission have agreed to reveal their identity: Andrew Azman (Reviewer #1); Sereina Herzog (Reviewer #2).

The reviewers have discussed the reviews with one another and the Reviewing Editor has drafted this decision to help you prepare a revised submission.

Summary

The paper by Larremore et al. presents a Bayesian framework to incorporate several sources of uncertainty (test characteristics, sample size, and heterogeneity across tested subpopulations) into estimates of SARS-CoV-2 seroprevalence and derived epidemiological/transmission parameters. They then use this framework to optimize study design and sampling schemes for serosurveys. While none of the methods presented are novel per se, the paper does present a much needed formal framework to guide the analysis and design of SARS-CoV-2 serosurveys. The paper was previously reviewed in another journal and the authors have provided the comments and responses. We thank them for providing these as they were very useful when evaluating the manuscript. We find that critiques have been adequately addressed.

Essential revisions:

1) As it is clear some research groups conducting large serosurveys have decided not to correct for assay performance, I think the authors missed an important opportunity to clarify the expected magnitude of biases (and false precision) from this. It would be very useful to put this in comparison to the biases and uncertainty expected from sampling and sampling of non-representative demographic samples.

2) It isn't clear why the authors chose to focus on 90% credible intervals as opposed to more commonly used 95% intervals. I don't think this is a major issue but seems like one of many decisions made throughout the paper that make these interesting results one (seemingly unnecessary) step removed from more practical application.

3) Figure 5: I don't see any attempt to explain why the modes for seroprev. posteriors are fairly off from the true values. Can you expand a bit and perhaps provide some solutions on how to do better?

4) Figure S4: Useful lessons here to illustrate how large samples from sub-populations alone (e.g., newborns and blood donors) can artificially increase apparent precision in seroprevalence estimates. Perhaps I missed this somewhere in the text but this seems useful to note.

5) The serological assays modelled in the text are not typical ones used in practice today. I appreciate that this manuscript was drafted early on the pandemic but the supplementary table should at least be updated to include the characteristics of some of the main assays in use today (e.g., EuroImmun, Roche, Abbot) for comparison purposes.

6) Some of the parameter assumptions cite other modelling papers as a reference. While some of these previous papers may have been based on empirical data, others are not. Please cite references for empirical data (e.g., incubation period) otherwise just be clear that these parameters were assumed.

---

## [Author Response]

Essential revisions:1) As it is clear some research groups conducting large serosurveys have decided not to correct for assay performance, I think the authors missed an important opportunity to clarify the expected magnitude of biases (and false precision) from this. It would be very useful to put this in comparison to the biases and uncertainty expected from sampling and sampling of non-representative demographic samples.

We now address this suggestion in two ways. First the text of the Introduction, Results, and Discussion, we note that a failure to adjust for test sensitivity and specificity will introduce bias. In numerical experiments there was no systematic way in which these biases of unrepresentative samples can be estimated without secondary data sources. However, we also note that biases from failure to adjust for sensitivity, specificity, and population demographics (via post-stratification) are avoidable, while non-representativeness in general is not.

Second, we highlight in Figure 4 where estimates based on uncorrected seroprevalence data would lie. This visually illustrates both the bias and false precision of using uncorrected estimates.

2) It isn't clear why the authors chose to focus on 90% credible intervals as opposed to more commonly used 95% intervals. I don't think this is a major issue but seems like one of many decisions made throughout the paper that make these interesting results one (seemingly unnecessary) step removed from more practical application.

This is a great suggestion. All plots and analyses have been reproduced using only 95% intervals.

3) Figure 5: I don't see any attempt to explain why the modes for seroprev. posteriors are fairly off from the true values. Can you expand a bit and perhaps provide some solutions on how to do better?

Figure 5 demonstrates the analysis and forecasting pipeline described in Figure 1 for a single stochastic simulation of serological test outcomes for each of the 4 sampling schemes. As a result, the estimates themselves are based on n=1000 samples spread across the individual age bins. For instance, even under uniform sampling, there are only 62-63 samples per bin. Consequently, re-running the generating code for Figure 5 using different random seeds may produce estimates that over- or under-estimate the ground-truth seroprevalence (used to stochastically generate the data) and the true model R_eff_. We would absolutely expect this for the uniform samples scheme.

However, for the other three sampling schemes, age groups with higher true seroprevalence are more likely to be sampled than those with lower seroprevalence. As a result, when information is shared across age groups, such that data from well sampled age groups inform the estimates for less sampled age groups, we expect a bias in estimates for the less sampled groups, which have systematically lower prevalence. Unfortunately, estimator bias due to unsampled or undersampled groups is not, itself, estimable a priori.

In summary, through the Bayesian modeling, we suffer the possible introduction of bias and in return obtain large reductions in uncertainty for less sampled groups. We now call this out in both the text and caption.

4) Figure S4: Useful lessons here to illustrate how large samples from sub-populations alone (e.g., newborns and blood donors) can artificially increase apparent precision in seroprevalence estimates. Perhaps I missed this somewhere in the text but this seems useful to note.

We now elevate this point explicitly in both the main text and the Discussion.

5) The serological assays modelled in the text are not typical ones used in practice today. I appreciate that this manuscript was drafted early on the pandemic but the supplementary table should at least be updated to include the characteristics of some of the main assays in use today (e.g., EuroImmun, Roche, Abbot) for comparison purposes.

We have now shifted the main text to focus on the Roche Spike IgG test, while the supplemental materials show EuroImmun and Abbott Architect data. Per FDA filings^^, these tests have characteristics of:

EuroImmun IgG, sensitivity 90% (27/30), specificity >99.9% (80/80)`

Roche IgG, sensitivity 96.6% (225/233), specificity >99.9% (5990/5991)

Abbott Architect IgG, sensitivity >99.9% (88/88), specificity 99.6% (1066/1070)

Analyses redone in both main text and supplement to reflect commonly marketed tests. References to tests now shifted to FDA filings from EuroImmun, Roche, and Abbott.

6) Some of the parameter assumptions cite other modelling papers as a reference. While some of these previous papers may have been based on empirical data, others are not. Please cite references for empirical data (e.g., incubation period) otherwise just be clear that these parameters were assumed.

Citations updated in Supplementary file 1 and original data sources cited where possible (rather than the modeling papers that used the original sources).

[Editors' note: we include below the reviews that the authors received from another journal, along with the authors’ responses.]

Reviewer #1 (initial submission):This manuscript discussed a framework to design serological surveys to estimate SARS-CoV-2 seroprevalence and epidemiological parameters by integrating serological data into epidemic models. The manuscript is well written, and the details of epidemic simulations and inferences are well documented. Please see below for my comments and suggestions:

We thank the reviewer for this appraisal of the manuscript.

Major comments:1) It is not clear in the manuscript whether the design of the serological surveys is one-off cross-sectional or serial cross-sectional at different time points. I think the underlying assumption is that the seroprevalence before the COVID-19 pandemic is zero. But practically if any serological survey is conducted now, it would require at least sampling at two time points so that infection attack rates could be estimated between these two timepoints, e.g. the recent serological survey in Geneva (Stringhini et al., 2020). Could the authors clarify their assumptions used in the simulation models and discuss if serial sampling could reduce the uncertainty in seroprevalence estimation?

Thank you for raising this issue. These assumptions have now been clarified in the Introduction and in the Discussion, where we explicitly note that our analysis is for the design of individual cross-sectional serosurveys. However, it is also the case that, given the results of an existing age-stratified serological survey, our framework would help to design a second survey which incorporated the information in the first. We now described exactly this procedure in the Discussion.

We also considered including the concept of a model in which the seroprevalence estimates are linked from one time point to the next, such that the successive waves of the survey could build on each other by creating a larger effective sample size. For instance, if one could justifiably assume that seroprevalence should always be increasing over time, then this information could be productively included in a statistical analysis. However, the evidence around the duration of positive antibody responses for SARS-CoV-2 remains mixed (for instance, see Herzog et al. Figures 2A-C ) and for now, we prefer not to include such strong monotonicity assumptions.

Separately, we have updated the citation for the Stringhini et al. paper to its now-published version, and note that our remake of Figure 5 (see reviewer #2 suggestions) uses empirical estimates from that paper to better ground our work in realistic potential outcomes.

2) It seems the time between infection and seropositivity was not considered in the simulation model (Table S2 if the recovery rate is 0.5). It takes 7-21 days for antibodies to develop to detectable levels (depending on whether IgM or IgG are antibody markers of interest). Thus, the inference about peak time and peak incidence presented in Figure 4 are difficult to interpret without the timing of sampling after considering the time between infection and development of antibodies.

We now address this idea two ways. First, we have rewritten parts of the manuscript to clarify that we are not inferring peak timing and height, but rather, we are using the serosurvey results—with uncertainties—as the initial conditions (via the depletion of susceptibles) from which forward-looking simulations could be run.

Second, we now discuss the fact that, due to delays in the accumulation of IgM/IgG, modeling studies that incorporate seroprevalence estimates should acknowledge such potential underestimates in interpreting their findings. This issue is inherited by all serological studies.

3) The MDI strategy presented in Figure 5 is interesting. But as mentioned earlier, the timing of sampling is not clear in the model assumptions. Since the authors have included age-specific susceptibility and POLYMOD-type contact matrices in the model structure, it is expected that the susceptibles deplete at different rates for different population subgroups stratified by age and susceptibility. This will inevitably affect the next generation matrix at different time points as the pandemic unfolds. How sensitive are the four sampling strategies to the assumption of the timing of sampling?

When age-stratified seroprevalence estimates are available, differences in susceptible depletion rates are accounted for in the calculation of R_eff_, as well as in the MDI recommendation, due to the term (I-D_θ_) in equation S8. In essence, the fact that prior information about contact structure and depletion of susceptibles is included in both the sample recommendation (MDI) and the eventual calculation (here, R_eff_) means that MDI is able to produce better estimates of R_eff_ and other model-based calculations.

The effect of the (I-D_θ_) term is that the next-generation matrix affects fewer and fewer individuals, on a per-subpopulation basis, as the epidemic continues, because fewer and fewer are susceptible. Therefore, when subpopulation seroprevalence estimates are available, mid-epidemic, the MDI recommendation is to oversample subpopulations that are critical for the ongoing dynamics and those subpopulations that are likely to have higher variance estimates due to their increased seroprevalence. These points are now emphasized in the Materials and methods, and have been further expanded in Supplementary Text S3.

4) Following the comment above, would the estimation of next generation matrix from seroprevalence is delayed considering the time between infection and seropositivity? How would this delay affect the R_eff_ estimation and the four sampling strategies in Figure 5?

We take this question to ask whether one could consider seroprevalence estimates taken at a particular time (or more realistically, over a particularly set of weeks), and then perform a sort of forward-adjustment for the fact that (1) individuals deemed seronegative at the time of sampling may now show robust antibody responses, post-convalescence, and (2) the dynamics of disease spread have continued to deplete susceptibles. In other words, could one adjust the estimate of R_eff_ to account for the delay between sampling and the calculation?

We believe that the answer to this question is yes, but to do so would require a model of disease dynamics and the ability to specify initial conditions from serological data. In other words, rather than estimating R_eff_ at the time of sampling, one could run the model forward by a few weeks and then estimate R_eff_ at that point instead. The uncertainty in the sampling-time estimates should therefore produce a cone of possible future-time estimates—precisely the type of modeling possible through the present work!

Minor comments:1) Could the colour gradients be replaced by lines or contour plots for Figure 2A and Figure 3B? The current version is a little difficult to read.

Great suggestion. We have made both into contour plots as suggested, as well as similar plots in Figures S1 and S2. We have also changed the shading in histograms for Figure 5 to be more legible.

Reviewer #1 (comments on revision):The authors have addressed all my comments.Reviewer #2 (initial submission):"Estimating SARS-CoV-2 seroprevalence and epidemiological parameters with uncertainty from serological surveys" describes a mathematical framework that addresses uncertainty from multiple sources (test characteristics, sample size, heterogeneity). Such a framework allows the final modelled forecasts to admit uncertainty estimates in the forward direction, or alternatively allows test design and allocations to be tailored to fit desired uncertainty tolerances in the reverse direction (Figure 1). The availability of such frameworks is highly desirable, given the potential impact of national-level policy interventions based on such modelling (e.g. citation [12],which contributed to the U.K. lockdown response)The presented framework considers three sources of uncertainty: imperfect sensitivity/specificity of test results, nonrepresentative sampled populations, and uncertain relationship between seropositivity and immunity. Overall, the derivation of the modelling as presented in the supplementary material appears to be sound, and the various explanatory examples demonstrating simulated outcomes over different parameters (e.g. subpopulations, overall seroprevalence, sampling strategies, test kits, etc) are fairly comprehensive. Given the potential far-reaching impact of accurate modelling of seroprevalence on public health (including uncertainty estimates), this manuscript would be a timely addition to the literature.

We thank the reviewer for this positive assessment.

There may however remain some issues for the authors' consideration:1) While it is noted at the end of the Introduction section that "…this framework can be used in conjunction with any model", it appears that the framework is broadly applicable to most types of epidemiological modelling tasks in general (not limited to SARS-CoV-2), since the integrated sources of uncertainty are usually present in such models. As such, the authors might consider emphasizing both the generalizable nature of the framework, as well as the chosen application (SARS-CoV-2), in the title.

The reviewer is correct, of course, that this work is a more general framework with applications beyond SARS-CoV-2. We now emphasize the generality of our work in the Introduction and Discussion, and, if approved by the Editors, suggest that perhaps a new title could simply omit SARS-CoV-2: “Estimating seroprevalence and epidemiological parameters with uncertainty from serological surveys”

2) The description of the method refers to subpopulations in general (e.g. Figure 1), although it seems that the subpopulations explored throughout the paper refer to age-based subpopulations (in particular, 5-year age bins). Moreover, while several real-life age distributions(binned age-i histograms) were explored (as described in the Data Sources section), the same seropositivity values (theta_i_) were assumed for each age-i subpopulation (Table S2), for all age distributions.While Bayesian hierarchical model sampling was deemed to produce sufficient estimates of the posterior distribution (S1.3/Figure 5), it might be noted that the various sampling strategies were evaluated solely on (age-based) subpopulations with relatively low variance in subpopulation seroprevalance (absolute difference capped at 2%, for an absolute average overall seroprevalance of about 10%). This might not be the case for other types of subpopulations; there may possibly be significant differences in seropositivity between subpopulations defined on other factors (e.g. geographical, race/ethnicity, etc). The authors might consider discussing/evaluating their various sampling strategies under circumstances of higher variance in subpopulation seroprevalence.

This point is well taken. Indeed, there can be far larger variation, particularly if estimates are stratified by other characteristics. Studies from New York City , Karachi , and Mumbai (where positive test rates were 54% and 16% in slums and non-slums, respectively), have all found substantial heterogeneities by neighborhood which exceed the heterogeneities assumed in our simulations, as well as the heterogeneities by age found in Swiss (Stringhini et al) and Belgian (Herzog et al) serosurveys.

We now discuss this point in the Materials and methods and Discussion sections of the paper in two ways. First, when there are large sample sizes across heterogeneous subpopulations, the parameters of the hyperprior will be overwhelmed by the data, and thus heterogeneity will not be an issue. Second, when sample sizes are low and heterogeneity is high, the hyperprior parameters for the variance between subpopulations can be adjusted to better accommodate larger variation between subpopulations. Even in these situations, the use of a hyperprior to share information across sparsely sampled (or even unsampled) bins is generally preferred.

3) Further, the modelling of seropositivity values (theta_i_) used throughout for simulation does not appear to have been justified with actual reported data (as for the contact matrices C_ij_). In particular, its modelling as an absolute deviance from the average seroprevalence seems unlikely to properly reflect actual seroprevalence distributions (e.g. "Seroprevalence of anti-SARS-CoV-2 IgG antibodies in Geneva, Switzerland (SEROCoV-POP): a population-based study", Stringhini et al., (2020) suggests significantly lower seroprevalences for the very young and the elderly), particularly when average seropositivity is low (and would be problematic when average seropositivity below 0.014, since this would produce negative values for some subpopulations). The authors might consider justifying the seropositivity value parameter modelling in greater detail, ideally with reference to published work.

To address this suggestion, we have now conducted an analysis identical to the previous Figure 5, but using age-stratified seroprevalence estimates as reported by Stringhini et al., (2020) as suggested. We used values reported therein for only seropositive and seronegative samples (Table 1), but discarded indeterminate results. We re-estimated seroprevalence values by age directly from test results as posterior means, using the sensitivity and specificity reported by the authors. This new analysis is shown in the remade version of Figure 5.

Because our submitted manuscript was developed prior to the publication of high-quality age-stratified analyses, we did not base our synthetic seroprevalences θi in real-data estimates.

However, since that time, multiple high-quality studies have been conducted, including the referenced Lancet study by Stringhini et al. All studies suggest more variation than we assumed in our synthetic seroprevalence estimates, though both values and trends by age vary, as summarized below:

19.7% (75y+) to 29.9% (0-17y)

0.9% (5-9y) to 10.8% (20-49y) [MLE from Table 1 values]

1.8% (5-9y) to 10.8% (20-49y) [Bayesian posterior mean with uniform prior from Table 1 values]

1.4% (20-30y) to 5.8% (0-10y)

3.8% (60-70y) to 15% (90y+)

3.7% (60-70y) to 11.1% (0-10y)

2.0% (60-70y) to 7.5% (10-30y)

2.1% (60-80y) to 9.0% (10-20y)

More broadly, the generation of synthetic and realistically varying age-stratified seroprevalence values for simulations is a challenge, due to the need to include variation that scales realistically with overall seroprevalence. Multiplicative scaling between age groups is a reasonable approach to generate synthetic values for lower seroprevalences (e.g. variation from 1% to 5% scales up to variation from 2% to 10%), but is unlikely to be reasonable at higher values. Because our numerical experiments scaled mean seroprevalence values from a minimum of 5% to a maximum of 50%, we therefore chose the variation to be additive.

As an aside, we note that the total range in the synthetic seroprevalence values ranged from mean–0.014 to mean+0.02, and the minimum value of the mean was chosen to be 0.05. We would therefore like to reassure the reviewer that no negative values were nonsensically drawn. However, we also feel that our explanation of these values and how they were used was unclear in our submission, and have therefore updated the manuscript (and Table S2).

4) Still on the seroprevalence modelling, the formulation of theta_i_ as theta~+ [-0.014, -0.012.… -0.012] would appear possibly inconsistent, depending on the actual subpopulation distributions used. For example, consider a (young) population that is evenly split between the first two age bins {0-4, 5-9} only. Then, the average seroprevalence derived from the definition of theta_i_ on these subpopulations would appear to be (1 – 0.013)theta^~^, which contradicts the initial definition of theta^~^ being the average seroprevalence.

This is an excellent point, and one which can easily be clarified. The synthetic subpopulation seroprevalences were chosen solely for illustrative purposes to demonstrate the performance of the method at various overall seroprevalence values. We now clarify this in the structure of Table S2, which has been split into values use in dynamical models (e.g. POLYMOD contact matrices) and values used as synthetic test parameters (e.g. values of n and θ).

Further, we note that θ*^~^* corresponds to the unweighted average of subpopulation seroprevalences. This reflects a modeling choice: each θii is drawn from a Β distribution with the same mean, and during inference, that mean is estimated as the unweighted average across subpopulation means. In this way, all subpopulations have an equal influence on the floating mean of the Β prior. However, to reassure a skeptical reviewer, we point out that the difference between the demography-weighted average and the unweighted average in the synthetic example of the manuscript is a seroprevalence difference of 0.001 (i.e. an unweighted average of 0.5 corresponds to a weighted average of 0.501).

A few minor suggestions follow:5) While SEIR models are explored, the authors might consider briefly discussing the SEIRSextension, given recent indications that antibody immunity to COVID-19 possibly wanes over months.

In light of this suggestion, and the suggestions of reviewer 1, we now discuss this point in the manuscript, inclusive of various immunological scenarios including partial protection, waning protection, and complete and durable protection (e.g. as in Saad et al.). Critically, however, we emphasize that our framework should work with forward simulations from any model, including S[E]IRS models and stratified agent-based models.

As a point of interest, we also note that whether or not protection wanes may be separate from whether or not seropositivity wanes, further complicating modeling more broadly.

6) Backtesting the proposed model with actual real-world data (vs. simulated cases), and comparing the predictions to actual observed trajectories, would appear to best illustrate the capabilities of the model. However, it is recognized that this may not be feasible.

We agree that this would be interesting, but also agree about feasibility. Indeed, our goal with this paper was to demonstrate how models, generically defined, could be used in both forward simulation and for serosurvey design, rather than to validate or calibrate any particular model.

7) Citation [20] seems to have just been published in Nature Medicine, and might be updated in the References section.

We have updated this citation, along with citations to other now-published works.

Reviewer #2 (comments on revision):We thank the authors for addressing our major concerns from the previous review round, in particular the inclusion of real-life survey data with higher age-group variances in the example analyses. There are no further comments.Reviewer #3 (initial submission):The article "Estimating SARS-CoV-2 seroprevalence and epidemiological parameters with uncertainty from serological surveys" by Larremore and colleagues presents a Bayesian hierarchical and modeling framework for estimating important epidemic quantities using seroprevalence studies. One of the key issues the authors advance is that Bayesian frameworks allow for propagation of uncertainty, which can then be immediately used as an input into an epidemic model if the Bayesian and modeling frameworks are linked. The concept is appealing, but the article, like many articles that use Bayesian frameworks, couch important assumptions in statistical considerations that make these assumptions appear unimportant. This is problematic in the entire field, and allows Bayesian statisticians to tinker with priors that yield many variants of posteriors without grappling with the assumptions that go into the choice of priors.A few comments below, and some summary suggestions at the bottom:"We denote the posterior probability that the true population seroprevalence is equal to θ, given test outcome data X and test sensitivity and specificity characteristics, as Pr(θ|X)". This notation implies that the posterior distribution of θ depends on X only. Specific notation for all other inputs is important, as θ depends on multiple factors simultaneously.

We agree and have modified the notation for the posterior distribution to be Pr(θ|X,se,sp). These changes also affect the notation in Figure 1, to match.

"Because sample size and outcomes are included in X, and because test sensitivity and specificity are included in the calculations, this posterior distribution over θ appropriately handles uncertainty" – just because some elements are in the conditional portion of the probability expression does not mean that uncertainty is handled appropriately. This is an important step that needs more careful handling.

We agree, and have rephrased this text to be more precise about which sources of uncertainty our calculations include. Specifically, we now write, “By explicitly conditioning on the data and test characteristics, the posterior distribution over θ captures the uncertainty in seroprevalence due to limited sample sizes and an imperfect testing instrument.”

The heart of the methods, including all the important but unstated assumptions, are in the Appendix S1. There are several steps that are glossed over and important assumptions are not explicit or tested.Plugging in equation S2 into S1…that may be ok but there's something that requires more thinking about plugging in a single-test probability expression (S2) into a population binomial function (S1). I appreciate the authors being clear that S2 is a single test probability (not Pr(n+|theta, u, v)), but it calls into question whether or not it can be then substituted back into S1, since those are different quantities. More explicit derivation there would be important.

Equation (S2) and (S3) are standard in the testing literature. We have added a citation to Diggle (2011) where both (S2) and (S3) appear in slightly different notation. Given the marginal probability of a random individual testing positive in (S2), the only additional assumption in the binomial model in (S3) is that the test outcomes of the individuals in the sample are independent. We added a statement to emphasize that this independence assumption underlies (S1) and (S3), and point readers to an example in which samples were modeled as non-independent within households due to correlated exposure.

Then, all the assumptions that go into moving from S3 to S4 are glossed over. Why assume an uninformative prior on theta? Why not assume some function with density around Pr(n+)? I don't know what is the best prior here, but I don't think the author do, either, and those assumptions are important to make explicit and assess as they propagate through the analysis. Similarly with the assumptions around the use of the Β distribution. Why that distribution? What are the implications of using different heuristics to anchor the priors? What are the implicit assumptions in this algorithm? Statistical considerations and expedience are unsatisfactory explanations for issues with a lot of real-world biology that can be used for setting of priors.

We place an uninformative prior on θ because we do not wish to bias the prevalence estimates in any way. For example, in the current SARS-CoV-2 epidemic, limited testing early in the pandemic is believed to have missed numerous infections and therefore seroprevalence might be expected to be much larger than prevalence estimates based on virological tests. In this setting, placing a uniform prior on the seroprevalence can be viewed as a conservative prior, which will result in wider credible intervals for the seroprevalence estimate compared to what would be obtained from an informative, but potentially mis specified, prior.

We use a Β distribution for the mean seroprevalence parameter in the age-structured model since it is a distribution on the interval [0,1] which can be parameterized in terms of its mean and variance. Although different priors would result in different posteriors, our use of a diffuse prior and weakly informative hyperprior were selected to be conservative in the estimation, thereby producing wider credible intervals reflecting higher uncertainty. An exploration of the impacts of particular choices of prior and hyperprior for a particular seroprevalence study can be found in recent work by Gelman and Carpenter, to which we now refer readers in a new paragraph in the Discussion.

The process for subpopulations is not entirely clear. It seems the authors choose a single prior for all subpopulations, which is then updated with subpopulation-specific data. However, it is unclear how the common prior is selected. While this may seem trivial, it can have very large implications on the posterior. Assuming high-variance priors will result in high-variance posteriors, and vice versa. Without more careful attention to the choice of priors, the authors allow the statistical cloak of Bayesian analysis to disguise the central assumptions that drive the outcomes of such analyses.

The common prior was chosen for the age-specific seroprevalence estimates to construct a hierarchical model. Hierarchical models have the key advantage that they pool information across age groups in constructing the age-specific estimates. However, the common prior does not imply that we expect no heterogeneity across subpopulations. Indeed, we investigate scenarios with heterogeneous seroprevalence levels in the manuscript, as do others, including Stringhini et al., (2020), and Carpenter and Gelman (2020). As noted above, we now include discussion of prior distributions in the manuscript’s Discussion.

The integration of the posterior estimates into SEIR models leave important pieces out. Specifically, it appears samples from the posterior distribution were "initially placed into the "recovered" compartment of the model." The prevalence estimates should not be used to parametrize the initial conditions. Indeed, parameterizing the model so as to achieve the estimated prevalence – use theta as a calibration target – is central to effective use of seroprevalence data. After all, that prevalence is achieved at a moment in time, and reflects (with some caveats) the cumulative incidence of regional infections and transmission from the beginning of the epidemic up to that point. That leaves an important question of t_0_ for the epidemic, but those are the issues that modelers should grapple with in incorporating seroprevalence data into SEIR models.

We emphasize that our efforts to connect serological studies and modeling efforts is designed for forecasting and forward simulation, but not backward inference of past epidemiological parameters. To clarify this point, we now describe models as forward-simulations, to avoid confusion.

In summary, a few suggestions:The article would be better if it focused on just one of the two key steps – either the estimation of prevalence, or the integration of prevalence into models. I think the authors have more strength in the former, as their handling of the latter is relatively superficial.Choosing priors is not merely a statistical exercise. Even choosing weakly informative priors is a nevertheless a choice with important implications for the posterior distribution. Let alone choosing Β functions, despite their convenient properties.The choice of priors – from the very origins of Bayesian statistics – should take into account knowledge about the real world that is then updated. If there is truly no prior information, then frequentist statistics can be just as useful and there is no need for Bayesian frameworks.However, that is rarely the case, even in the case of the novel coronavirus. Priors about seroprevalence can be informed by factors such as case counts population density, demographic structures, and more.

As noted above, we now discuss this point directly, and frame the tradeoff between uninformative and informative priors in terms of being conservative about uncertainty vs justifying additional assumptions. While we understand that this review of a complicated topic is far from all there is to say on the matter, we hope that those interested in the topic of using orthogonal data sources to create informative priors will inform themselves appropriately.

A truly interesting paper would examine the sensitivity of Bayesian models to the choice of priors in estimating covid seroprevalence. I would be interested in reading that paper!

Indeed! We suggest Gelman and Carpenter, 2020, which investigates the statistically controversial Santa Clara seroprevalence study from Bendavid et al.

Reviewer #3 (comments on revision):I appreciate the opportunity to revisit the article "Estimating SARS-CoV-2 seroprevalence and epidemiological parameters with uncertainty from serological surveys" by Larremore and colleagues. The authors have revised the manuscript to include more age-structured data and a few minor text revisions (little of substance appears to have changed in the tracked manuscript).I appreciate the authors' clear exposition of their goals and the consistent thread of producing seroprevalence estimates through to their incorporation into forward-looking epidemiologic models.However, the authors continue to gloss over the hard issues with Bayesian approaches, and continue to present this approach as an improvement over existing approaches. For example, there are several places where the authors imply that their approach affords the "appropriate" propagation of uncertainty. Along a similar vein, the response to the choice of uninformative priors is described as Conservative. While the authors' intent in using the word Conservative is Wider confidence intervals, the choice of the word Conservative points to the likely-implicit biases of the authors which deserve more careful consideration.Choosing the priors – and the width of the posterior uncertainty – is a choice which has tradeoffs. The suggestion that this is a "Conservative" choice suggests the authors may think estimating too-wide credible/confidence intervals is preferable to estimating too-narrow intervals. However, this is a bias. In genetics, using Bonferroni-type "conservative" thresholds for associations leads to discarding many true discoveries that may have important implications for science and medicine. Similarly, choosing priors that imply relatively large uncertainty around SARS-CoV-2 (or any antibody) seroprevalence could have important implications for things like epidemic projections, with large consequences. Both under-estimating and over-estimating projected infections or other outcomes have downsides, and being more "conservative" (in the authors' sense) is not "appropriate" or preferable; rather, it is a choice. A close look at the tradeoffs involved would lend much greater credibility to the paper.An important related challenge is that Bayesian models – to date, given their relative infrequent use compared to frequentist statistics – obscure important assumptions and allow investigators degrees of freedom that are hard to identify. The authors reference a Gelman-Carpenter analysis that had done just that. Gelman and Carpenter chose one prior distribution, examined the posterior distribution, didn't like the findings, and Gelman noted "So I'll replace the weakly informative half-normal(0, 1) prior on σ_sens_ with something stronger: σ_sens_ ~ normal(0, 0.2)." This kind of investigator-dependent statistical sleight-of-hand is common in Bayesian statistics but is hard to identify (it is rarely so explicitly noted) and rarely examined with balance.Indeed, the authors correctly acknowledge that "specifying such priors appropriately relies on additional information and/or assumptions … which may be sparse, particularly during an unfolding pandemic." And this brings up the last point. When the prior information is sparse, frequentist approaches provide a more transparent and familiar framework for analysis. Both Bayesian and frequentist approaches have their place – and both have important strengths and limitations. And a novel phenomenon (such as COVID-19) with little prior information may well be the kind of situation where frequentist approaches are advantageous relative to Bayesian approaches. Indeed, nearly all COVID-19 seroprevalence studies have used frequentist statistical approaches exactly for that reason: so much is unknown about COVID-19, let us not add obfuscation to uncertainty. (As examples with COVID-19, see Anand, Lancet 2020; Pollàn, Lancet 2020; and Sood, JAMA 2020. For a review of seroprevalence studies, see Ioannidis, medRxiv doi 10.1101/2020.05.13.20101253.)In the end, I do not think Bayesian approaches are, to date, in a place where they provide an improvement over frequentist approaches for seroprevalence estimation (nor for providing inputs into SEIR models). On the other hand, a balanced comparison of their merits and limitations would go a long way to furthering the field.

First, the core problem solved in our work is not solvable using frequentist methods. Our manuscript is about the interplay between epidemiological modeling and the design and analysis of serosurveys. To fully connect the two requires inferences about seroprevalence and uncertainty to be done at the same level of age (or subpopulation) stratification. These inferences are not possible using frequentist methods, especially for the convenience samples like blood donations or neonatal dried blood spots which we develop and analyze in the paper, since the samples inherently do not contain individuals from all subpopulations. The Materials and methods in our paper, whose assumptions we have been entirely clear and transparent about, enable this connection. Indeed, the analysis methods of seroprevalence in the three research articles cited by Reviewer #3 are insufficient to accomplish this key task—and perplexingly, one does not even correct for the sensitivity and specificity of the tests (Pollán et al.).

Second, Bayesian methods are widely and increasingly used precisely because of their flexibility to provide transparent inference in contexts where traditional frequentist methods fall short. Indeed, many journals, including this one, have published many high-impact papers based on Bayesian analyses, on a variety of topics including H1N1, malaria, depression, and infertility. In fact, the landmark seroprevalence study by Stringhini et al. which the other Reviewers noted and on which we based our revised analysis of real-world data, is, itself, a clear, high-impact example of a Bayesian analysis in the context of SARS-CoV-2 seroprevalence. Reviewer #3 takes issues with Bayesian methods in general; however, the reviewer’s opinion is out of step with the broader literature, past publications in infectious disease epidemiology and modeling, and mathematical modeling and inference under uncertainty.

In sum, although reviewers #1 and #2 approved of our manuscript—and indeed improved it with constructive comments which called for new and more detailed analyses of real data—Reviewer #3 is narrowly focused on the presence of Bayesian inference methods in the manuscript, full stop, and has provided no clear, constructive suggestions. It is our view that this has led to the reviewer’s very targeted dismissal of the paper without considering its goals.

We are not interested in re-litigating an imagined dispute between Bayesians and frequentists, and instead appeal to “the recognition that each approach has a great deal to contribute to statistical practice and each is actually essential for full development of the other approach.” We are interested in the topic on which our manuscript is focused: bridging the divide between serological studies and epidemiological modeling, for the improvement of both.